

# 1 An Operational Thermodynamic-Dynamic Model for the Coastal

# 2 Labrador Sea Ice Melt Season

Ian D. Turnbull[1], and Rocky S. Taylor[1,2]
[1]C-CORE
[2]Memorial University of Newfoundland (MUN)
*Corresponding author*: Ian Turnbull (ian.turnbull@card-arctic.ca)
**1 C-CORE**
**Captain Robert A. Bartlett Building**
**1 Morrissey Road**
**St. John's, Newfoundland and Labrador A1B 3X5 Canada**
**Ian Turnbull Phone: (709)-864-6208**
**Fax: (709)-864-4706**
**2 Memorial University of Newfoundland**
**Faculty of Engineering and Applied Science**
**S.J. Carew Building**
**240 Prince Philip Drive**
**St. John's, Newfoundland and Labrador A1B 3X5 Canada**
**Rocky Taylor Phone: (709)-864-4370**
**Fax: (709)-864-4706**
**E-mail: rocky.taylor@card-arctic.ca**



**Abstract**

An offshore operations thermodynamic-dynamic prediction model of sea ice break-up and drift is presented for central coastal Labrador in Atlantic Canada, and demonstrated for portions of the 2015 spring break-up of the land-fast ice. The model validation is performed using the data from ice tracking buoys deployed on the land-fast ice, which began drifting after break-up of the land-fast ice. The model uses a one-dimensional thermodynamic parameterization for ice melt and growth, includes snow accumulation and melt, and melt-pond and lead growth and contraction. The dynamic model uses a Smoothed Particle Hydrodynamics (SPH) parameterization for ice motion and changes in ice thickness and concentration. The dynamic forcing parameters include wind and ocean current drag, Coriolis deflection, internal ice stresses, and gravitational forcing due to sea surface gradients. A coastal repulsion force is employed to prevent ice particles from crossing the coastal boundaries. The model is sensitive to the prescribed initial snow depth on the sea ice. In the present work, analysis of results is focused on the offshore regions of Makkovik and Nain, Labrador. The melt of the coastal land-fast ice in these regions can be adequately simulated by the thermodynamic model alone. The model predicts the timing of the local land-fast ice break-up to within 4.6 hours to six days, and can simulate observed ice buoy drift speeds to within 1.5 meters per second.

**1 Introduction**

Sea ice forecast models are an important tool for offshore industries operating in ice-prone environments, such as those involved in hydrocarbon exploration and development or marine transport. Accurate pack ice model forecasts allow for operations to be planned for maximum safety, efficiency, and minimum downtime as operators make the best possible preparations for the anticipated ice conditions, and can develop an advance understanding of how those conditions and associated hazards are likely to evolve during operations. While small-scale models are useful for generating drift forecasts for single or several ice floes over spatial scales of up to tens of km, regional and basin-scale models provide forecasts of ice conditions on scales of tens to hundreds of km, and permit operators to develop a picture of changes in ice conditions over such a broad area in which vessels and drilling platforms may travel. Such forecast models are useful for generating ice forecasts on timescales of days to months.

In this paper, a model is presented to simulate the melt and break-up of the land-fast ice along the central Labrador coast during the spring break-up season, as well as the regional ice dynamics in the further offshore marginal ice zone (MIZ). The model is designed to be an operational planning and ice forecast tool for the offshore oil and gas industry, as well as the marine transport industry. It is designed to accept gridded metocean input parameters from numerical weather prediction and ocean models. The model simulations presented here are run for selected periods during April-May 2015, and are compared to in-situ observations of ice drift as recorded by tracking buoys deployed on ice floes, and to reanalysis data on ice thickness, concentration, and velocity as provided by the Operational Mercator global Ocean analysis and forecast system. The models runs are forced with atmospheric and precipitation reanalysis data from the North American Regional Reanalysis (NARR), the European Center for Medium Range Weather Forecasting (ECMWF), and with ocean model output from the Operational Mercator global Ocean analysis and forecast system.



The results presented here focus particularly on the regions offshore Makkovik and Nain, Labrador, as these
locations have been identified as economically important for vessel transit. Makkovik has been identified as a
potential landing site for a pipeline from oil and gas fields offshore Labrador, and Nain is located near the port of
Edward's Cove which serves the Voisey's Bay nickel mine (Vale, 2015). In this paper, two locations offshore
Makkovik and Nain are selected to demonstrate the thermodynamic model simulation of the melt and break-up of
the land-fast ice during April 1 – May 31, 2015. Results from the thermodynamic model are also presented for the
central coastal Labrador region from Makkovik to Nain. Subsequently, the coupled thermodynamic-dynamic model
is demonstrated for the Makkovik-Nain region for May 1-7, 2015, which coincides with the break-up of the land-fast
ice offshore Nain.

### 1.1 Ice tracking buoy deployments

On April 9, 2015, six Iridium satellite-tracked buoys were deployed toward the outer edge of the land-fast ice
offshore Labrador (see Figure 1). The buoys were dropped out of a Twin Otter fixed-wing airplane, and were
deployed in two triangular arrays offshore Makkovik and Nain. These buoy deployments served to collect data on
the precise timing of the land-fast ice break-up at these locations during the 2015 melt season, and the subsequent
drift patterns of the broken floes. Table 1 summarizes the offshore deployment areas for the six buoys and the time
periods over which they drifted on ice following the break-up of the land-fast ice on which they were deployed. The
on-ice drift periods were determined from analysis of the internal temperatures transmitted by the buoys, as the on-
ice periods were characterized by distinctly larger diurnal temperature fluctuations in response to air temperatures,
compared with more homogeneous temperature records once the buoys fell into open water. The open water drift of
the buoys is not considered in this paper.
Buoys 1, 4, and 6 were deployed offshore Nain, and buoys 2, 3, and 5 were dropped offshore Makkovik. The first
drift motion recorded by the buoys took place on April 23 offshore Makkovik, and on May 1 offshore Nain. Buoy 5
failed on May 28 on the land-fast ice offshore Makkovik and never recorded any movement (Table 1).



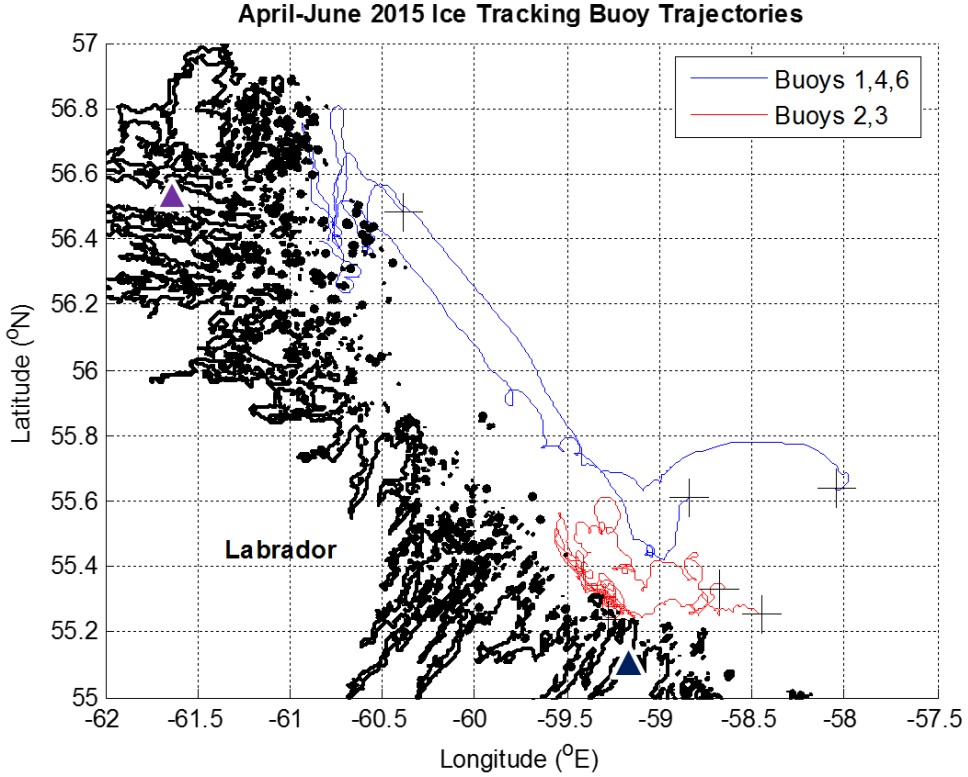


**Figure 1.** Trajectories of six ice tracking buoys offshore Labrador during April 23 − June 12, 2015. The dark blue
and purple white-outlined triangles mark the locations of Makkovik and Nain, respectively. The black plus (+)
symbols mark the ends of the buoys' on-ice drift trajectories.

**Table 1.** Deployment areas and time periods over which the six ice buoys drifted on ice following the break-up of
the land-fast ice.

| Buoy | Deployment Area | Ice Drift Period |
|---|---|---|
| 1 | Nain | May 1-18 |
| 2 | Makkovik | April 23 – June 12 |
| 3 | Makkovik | April 23 - June 8 |
| 4 | Nain | May 2-11 |
| 5 | Makkovik | N/A |
| 6 | Nain | May 6-19 |

**1.2 Regional environment**





Coastal Labrador is dominated by the Labrador Current, which has a mean surface speed of 0.25-0.5 m s$^{-1}$ (Lazier
and Wright, 1993) toward the south and SSE, parallel to the coast (Figure 2).

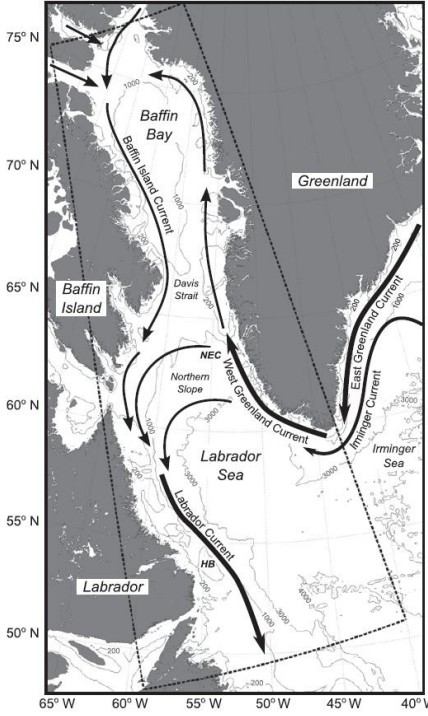


**Figure 2.** Mean surface currents offshore Labrador flow parallel to the coast toward the south and SSE at speeds of
0.25-0.5 m s$^{-1}$ (image reproduced from Fenty and Heimbach, 2013).
Freeze-up along the Labrador coast typically begins in December, with land-fast ice forming from the northern edge
of Labrador to just south of Lake Melville (Canadian Coast Guard, 2013). By April 1$^{st}$, a large region of 9+/10ths
mainly first-year (FY) ice has formed further offshore from the land-fast ice and drifts southward with the Labrador
Current, being replenished by more northerly-sourced ice from Baffin Bay (Figure 3). The boundary between the
regions of 9+/10ths pack ice and open water is typically sharp and along the edge of the Labrador Shelf (see Figure
5) near the western edge of the Labrador Current (Yao et al., 2000b), with only a thin band of ice 1-8/10ths in
concentration along the boundary (Figure 3).



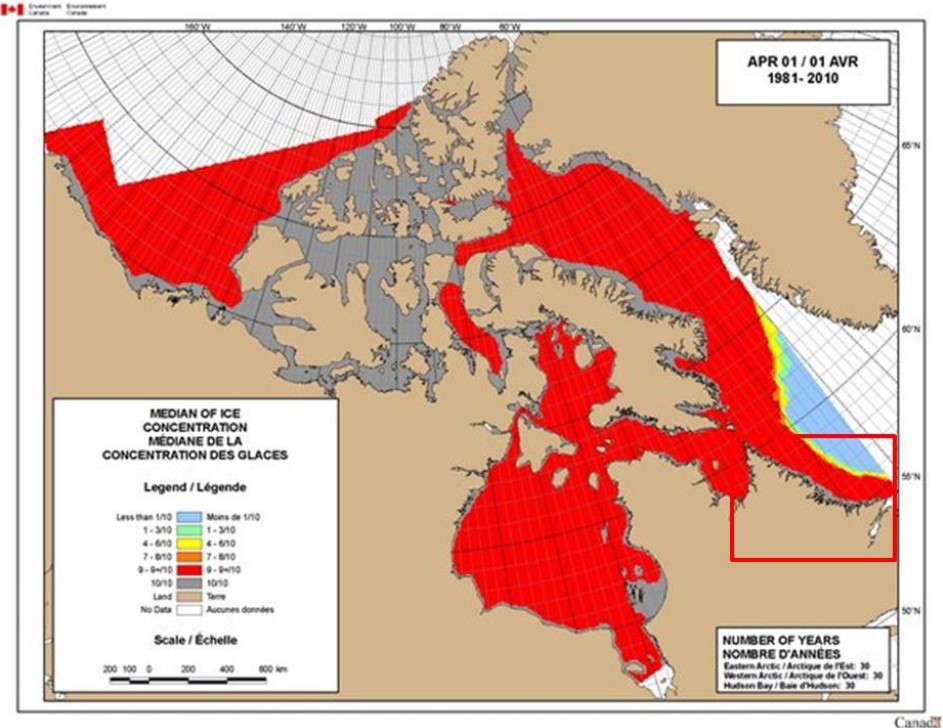


**Figure 3.** Maximum spring ice cover offshore Labrador (red-boxed region) typically includes a band of 10/10ths land-fast ice (gray region), and a more extensive zone of dynamic pack ice 9+/10ths in concentration (red region) further offshore (Canadian Coast Guard, 2013).

The onset of break-up of the high-concentration ice along the coast of Labrador typically occurs in mid-June for most of the region, with break-up commencing in early July for the most northerly areas (Figure 4).



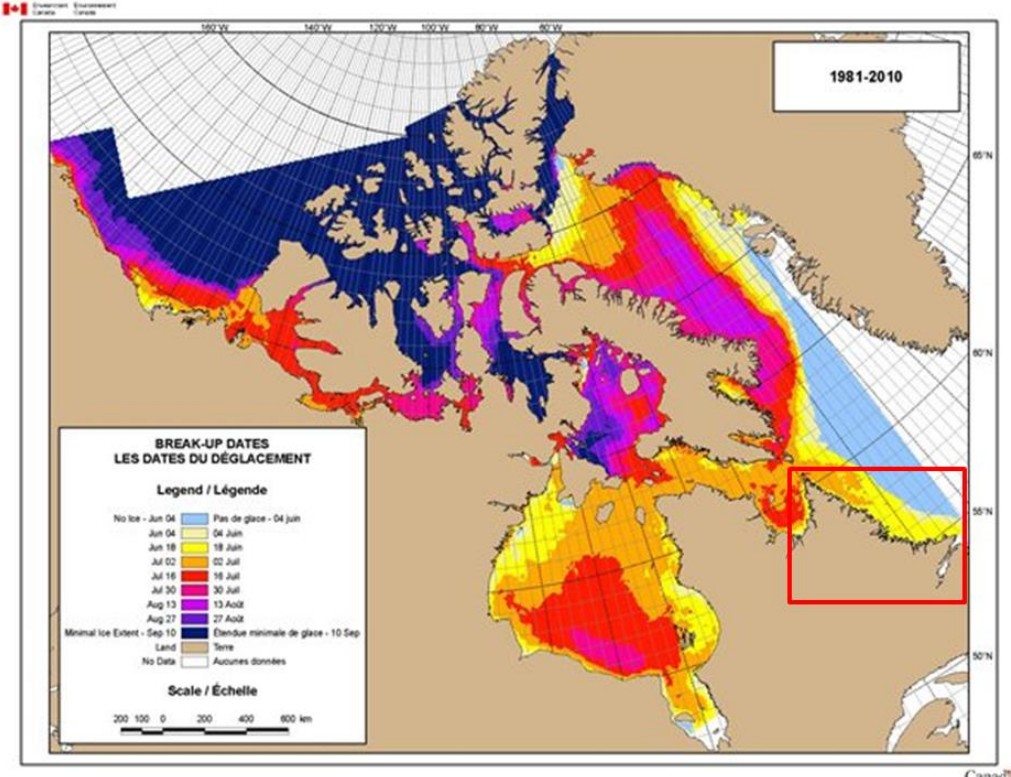


**Figure 4.** The onset of pack ice break-up in the offshore Labrador region (red-boxed region) typically occurs in mid-
June to early July (Canadian Coast Guard, 2013).
The water depth offshore Labrador extends to approximately 500 m along the Labrador Shelf, with a few pockets as
deep as about 1000 m (Figure 5). Further offshore, the depth rapidly increases to greater than 2000 m. At its
maximum seasonal extent in March-April, sea ice typically extends to the shelf break front (Yao et al., 2000b) along
the 500 m depth contour.



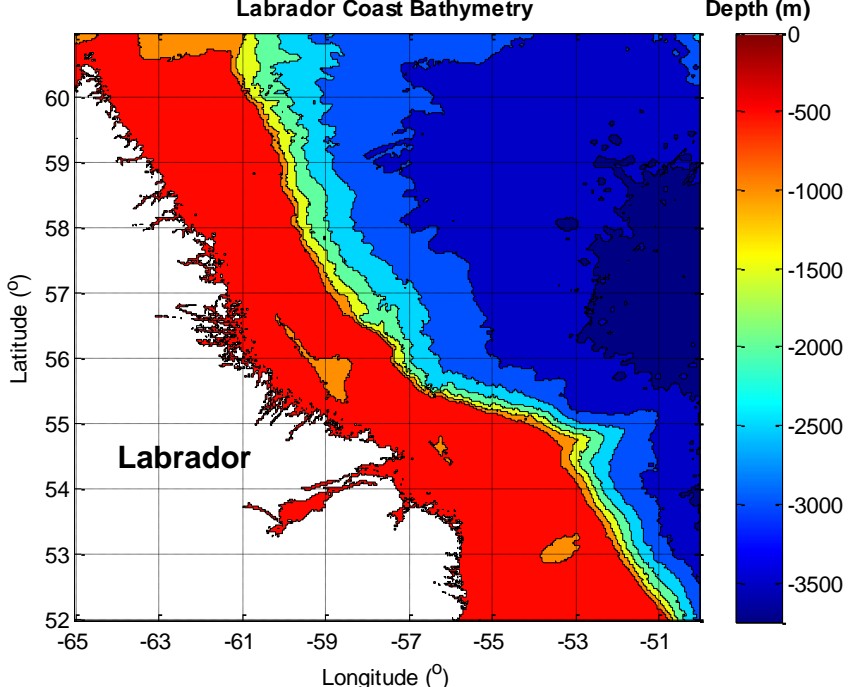

**Figure 5.** The bathymetry offshore Labrador (General Bathymetric Chart of the Oceans, GEBCO, 2008, 30-second data).

## 2 Ice Environmental Modeling

### 2.1 Previous modeling studies

Existing modelling studies for the offshore Labrador ice environment tend to focus on seasonal to inter-annual sea ice variability and its interaction with oceanic mixing and convection. Few studies have presented efforts to develop operational ice forecast models for the region.

A study by Peng, 1989, for example, applies a coupled ice-ocean model to simulate ice growth and transport in the Labrador Sea over a single winter season.

Prinsenberg and Yao, 1999 presents a climatological study of the seasonal evolution of sea ice cover offshore eastern Canada for 1991-1992 using a three-dimensional coupled ice-ocean model. The study additionally presents model simulations of ice cover under a scenario of a doubling of atmospheric carbon dioxide concentration by using as input the expected atmospheric conditions under such a scenario.





A paper by Yao, 2000 focuses on the assimilation of satellite-derived sea surface temperature (SST) data into a
coupled ice-ocean model for the Labrador Sea, with the primary objective of improving the modelled location of the
ice edge. The SST data assimilation, however, did not improve model performance in this regard.
A study by Yao et al., 2000a focuses on verification of a coupled ice-ocean model for the Newfoundland shelf
during February-April 1997, with the region of interest (ROI) stretching from offshore central-northern
Newfoundland to southern Labrador. The model output was compared in particular to observed ice edge location,
southern ice extent, and ice thickness as derived from Canadian Ice Service (CIS) charts, and to ice drift velocities as
measured by satellite-tracked ice buoys.
Yao et al., 2000b presents results of a modelling study of the seasonal variation of sea ice in the Labrador Sea using
an ice model coupled to the Princeton ocean model. In this paper, the Princeton ocean model is forced by monthly
climatological atmospheric data.
A paper by Sayed et al., 2002 presents the operational ice dynamics model developed by CIS for ice forecasting in
regional environments. The model uses a Lagrangian Particle-In-Cell (PIC) approach to model ice advection. In the
paper, a seasonal simulation of the model is presented, with the ice model coupled to the Princeton ocean model.
The model output is compared with the measured trajectory of a single ice tracking beacon offshore Labrador during
January-April 1999, and with the January-May 1999 evolution of the total ice area and mean thickness over a large
area encompassing the east coast of Canada.
A study by Fenty and Heimbach, 2013 explores the relationship between sea ice and oceanic variability in the
Labrador Sea and Baffin Bay. It analyzes the response of the annual sea ice cover variability to ocean currents and
water mass mixing and convection in the region.
A paper by Cooke et al., 2014 presents a study of inter-annual sea ice variability in the western Labrador Sea using a
coupled sea ice-ocean model. The paper focuses on the relationships between sea ice concentration and the
underlying water column properties in terms of temperature, salinity, and convection. Cooke et al., 2014 use the
Nucleus for European Modelling of the Ocean (NEMO) model and couple it to the Louvain Sea Ice Model 2 (LIM2)
multi-layered sea ice model. Atmospheric forcing is provided by the National Center for Environmental
Prediction/National Center for Atmospheric Research (NCEP/NCAR) six-hourly reanalysis data from 1948-2005.

### 2.2 Thermodynamic model description

The model presented in this paper was developed in the programming language of MATLAB. The thermodynamic
model is largely based on the one-dimensional model presented in Ebert and Curry, 1993 (hereafter referred to as
EC93), which is in turn based on the model presented in Maykut and Untersteiner, 1971. The thermodynamic model
can be run as a standalone model, or can be coupled to the dynamic ice model. The thermodynamic model accounts
for heat fluxes from the surface ocean through the ice slab, heat exchanges with the surface air or snow layer if
present, snow layer accumulation, melt, and heat exchange with the ice and atmosphere, incoming shortwave (solar)
radiation, total cloud cover and its effects on atmospheric albedo and down-welling longwave radiation, and lead
and melt pond growth and contraction. The thermodynamic model-only runs are initialized with ice particles on a





grid with 0.45° spacing (approximately 50 km). Ice particles are dropped from the model grid once they have
completely melted away.

### 2.2.1 Ice and snow thermal properties and heat conduction

In the thermodynamic model, the sea ice vertical temperature profile and thickness are governed by the one-
dimensional heat equation as per EC93,
$$(\rho c)_i \left(\frac{\partial T_i}{\partial t}\right) = k_i \left(\frac{\partial^2 T_i}{\partial z^2}\right) = \frac{\partial F_c}{\partial z}, \tag{1}$$

where $(\rho c)_i$ is the volumetric heat capacity of ice, $T_i$ is ice temperature (K), $t$ is time (seconds), $k_i$ is the thermal
conductivity of ice, $F_c$ is the upward conductive flux (W m$^{-2}$), and $z$ is the depth level in the ice slab (m) with $z = 0$
defined at the ice surface. As the heat capacity and thermal conductivity of ice are affected by its temperature and
salinity, the volumetric heat capacity of the sea ice (J m$^{-3}$ K$^{-1}$) is defined as (*e.g.*, EC93),
$$(\rho c)_i = (\rho c)_{i,f} + \frac{\gamma S_i}{(T_i - 273)^2}, \tag{2}$$

where $(\rho c)_{i,f}$ is the volumetric heat capacity of pure ice ($1.883 \times 10^6$ J m$^{-3}$ K$^{-1}$) and $\gamma$ is a constant ($1.715 \times 10^7$ J K m$^{-3}$
ppt$^{-1}$), and $S_i$ is the ice salinity (ppt). The ice thermal conductivity (W m$^{-1}$ K$^{-1}$) is defined as (*e.g.*, EC93),
$$k_i = k_{i,f} + \frac{B S_i}{T_i - 273}, \tag{3}$$

where $k_{i,f}$ is the thermal conductivity of pure ice (2.034 W m$^{-1}$ K$^{-1}$), and B is a constant (0.1172 W m$^{-1}$ ppt$^{-1}$). As per
EC93, $T_i$ is constrained to values 272.9K or less in order to avoid a singularity at 273K in Eqs. (2) and (3). The ice
salinity (ppt) is approximated by the equation from Timco and Johnston, 2002 for cold FY ice as a function of
thickness, *e.g.*,
$S_i = 13.4 - 17.4 h_i$          for $h_i \leq 0.34$m,
$S_i = 8.0 - 1.62 h_i$          for $h_i > 0.34$m,     (4)
where $h_i$ is ice thickness (m).
The ice temperature profile is initialized with a linear trend between the boundary conditions set at the ice base and
surface. The ice basal temperature is fixed at the melting point of seawater, and the ice surface temperature is
calculated as a function of the surface (2m) air temperature. The ice temperature is calculated at four evenly-spaced
points in the vertical slab, including the boundaries at the ice-water and ice-snow or ice-air (when no snow is
present) interfaces.
The snow-free ice surface temperature is calculated as a function of the surface air temperature according to the
function (*e.g.*, Timco and Frederking, 1990),
$T_{i,s} = T_a$          for -2 ≥ $T_a$ ≥ -10,
$T_{i,s} = 0.6 T_a - 4$          for $T_a <$ -10,





$T_{i,s} = -2.0$        for $T_a > -2$,    (5)
where $T_{i,s}$ is the ice surface temperature (°C), and $T_a$ is the 2 m air temperature (°C). When the air temperature is
higher than -2°C, the ice surface temperature is assumed to be at the approximate freezing point of saline FY sea ice,
-2°C.
The basal ice temperature is calculated as the freezing temperature of seawater as a function of sea surface salinity
(*e.g.*, Fofonoff and Millard Jr., 1983),
$T_{o,f} = -5.33x10^{-7}S^3 - 9.37x10^{-6}S^2 - 0.0592S$,        (6)
where $T_{o,f}$ is the freezing temperature of seawater (°C), and $S$ is the sea surface salinity (ppt).
When a snow layer is present on the sea ice, the snow vertical temperature profile is governed by a similar one-
dimensional heat equation as that used for sea ice (*e.g.*, EC93),
$(\rho c)_s \left(\frac{\partial T_s}{\partial t}\right) = k_s \left(\frac{\partial^2 T_s}{\partial z^2}\right) = \frac{\partial F_c}{\partial z}$,        (7)
where $(\rho c)_s$ is the volumetric heat capacity of snow, $T_s$ is the snow temperature (K), $k_s$ is the thermal conductivity
of snow, and $z$ is again defined with $z = 0$ at the snow surface. The snow temperature is calculated at three evenly-
spaced points in the vertical slab, including the boundaries at the snow-ice and snow-air interfaces. The volumetric
heat capacity of snow is given by (*e.g.*, EC93),
$(\rho c)_s = \rho_s(92.88 + 7.364T_s)$,        (8)
where $\rho_s$ is the density of the snow layer (kg m$^{-3}$) on the sea ice. The value of $\rho_s$ is 330 kg m$^{-3}$ if the surface air
temperature is less than 0°C, and is 450 kg m$^{-3}$ if the surface air temperature is 0°C or greater (EC93). Compaction
of the snow layer due to fresh snowfall and time is neglected in the present model. The snow thermal conductivity is
given by (*e.g.*, EC93),
$k_s = 2.845 \times 10^{-6}\rho_s^2 + 2.7 \times 10^{-4} \times 2^{\frac{T_s-233}{5}}$.        (9)
The temperature at the snow-ice interface is computed from Eq. (7) and is set to the ice surface temperature. The
temperature at the snow-air interface is initialized in K by (*e.g.*, Tonboe et al., 2011),
$T_{s,top} = 1.14T_a - 37.94$        for $T_a < 273.15$,
$T_{s,top} = 273.15$        for $T_a \geq 273.15$.    (10)
Snow temperatures are constrained to 0°C and below. The effect of sea water flooding on the snow surface is
neglected in this model. As with the vertical ice temperature profiles, at the start of each model run, the vertical
snow temperatures are initialized with linear vertical profiles. From the second time-step forward, the temperature at
the snow or snow-free ice surface is solved using the net surface heat flux (see Eq. 49 in section 2.2.7).
**2.2.2 Solar radiation**





The incoming solar (shortwave) radiation (W m$^{-2}$) at the top of the atmosphere is given by (*e.g.*, FAO, 2015),
$$F_{SW} = \frac{12 S_o d_r \left( (\omega_2 - \omega_1) \sin(\phi) \sin(\lambda) + \cos(\phi) \cos(\lambda) (\sin(\omega_2) - \sin(\omega_1)) \right)}{\pi}, \qquad (11)$$
where $S_o$ is the solar constant (1365.5 W m$^{-2}$), $d_r$ is the inverse relative Earth-Sun distance (radians), $\omega_1$ and $\omega_2$ are
the solar time angles (radians) immediately before and after a given time-step, respectively, $\phi$ is the latitude
(radians), and $\lambda$ is the solar declination (radians). The inverse relative Earth-Sun distance is defined by (*e.g.*, FAO,

227   2015),

$$d_r = 1 + 0.033 \cos\left(\frac{2\pi J}{365}\right), \qquad (12)$$
where $J$ is the Julian date. The solar declination is defined by (*e.g.*, FAO, 2015),
$$\lambda = 0.409 \sin\left(\frac{2\pi J}{365} - 1.39\right). \qquad (13)$$
The solar time angles are given by (*e.g.*, FAO, 2015),
$$\omega_1 = \omega - \frac{\pi dt}{24},$$
$$\omega_2 = \omega + \frac{\pi dt}{24}, \qquad (14)$$
where $\omega$ is the solar time angle (radians) at the midpoint of the period over which the incoming solar radiation is
calculated, and $dt$ is the length of the calculation period (hours). The solar time angle at the midpoint of the period is
defined by (*e.g.*, FAO, 2015),
$$\omega = \frac{\pi \left( (t + 0.06667(L_z - L_m) + S_c) - 12 \right)}{12}, \qquad (15)$$
where $t$ is the standard clock time at the hour midpoint (*e.g.*, $t = 13.5$ if the calculation time is between 13:00 and
14:00 hours), $L_z$ is the longitude of the center of the local time zone (60°W for the Atlantic Standard Time or AST
zone) in degrees west of Greenwich, $L_m$ is the longitude of the calculation site in degrees west of Greenwich, and $S_c$
is the seasonal correction for solar time (hours). The solar time seasonal correction is given by (*e.g.*, FAO, 2015),
$$S_c = 0.1645 \sin(2b) - 0.1255 \cos(b) - 0.025 \sin(b), \qquad (16)$$
where $b$ is defined as $\frac{2\pi(J-81)}{364}$ (FAO, 2015).
Finally, the sunset hour angle $\omega_s$ (radians) is used to determine whether or not the sun is below the horizon at a
given time. The sunset hour angle is defined as (*e.g.*, FAO, 2015),
$$\omega_s = \text{acos}(-tan(\phi)\, tan(\lambda)). \qquad (17)$$
The sun is below the horizon when $\omega < -\omega_s$ or $\omega > \omega_s$ and $F_{SW} = 0$. Note that times used to calculate $F_{SW}$ are
local times for AST, which is UTC-3:00 during April-May.
The flux of solar radiation (W m$^{-2}$) which penetrates the snow-free ice surface as a function of depth in the ice, $z$
(m), is governed by Beer's Law (*e.g.*, EC93),





$$F_{i0}(z) = I_0(1 - \alpha_i)F_{SW}e^{-\kappa_i(z)}, \tag{18}$$

where $I_0$ is the fraction of solar radiation which penetrates the atmosphere, $\alpha_i$ is the ice albedo, and $\kappa_i$ is the bulk shortwave extinction coefficient for sea ice (1.5 m⁻¹). The fraction of solar radiation penetrating the atmosphere is defined by (*e.g.*, EC93),

$$I_0 = 0.18(1 - T_{cc}) + 0.35T_{cc}, \tag{19}$$

where $T_{cc}$ is the total fractional cloud cover. If there is snow on the sea ice, then Eq. (18) becomes,

$$F_{i0}(z) = (1 - \alpha_i)F_{i0,s}e^{-\kappa_i(z)}, \tag{20}$$

where $F_{i0,s}$ is the fraction of solar radiation which penetrates the snow layer. The fraction of solar radiation penetrating the snow layer is given by,

$$F_{i0,s} = I_0(1 - \alpha_s)F_{SW}e^{-\kappa_s(z)}, \tag{21}$$

where $\alpha_s$ is the snow albedo, $\kappa_s$ is the bulk shortwave extinction coefficient for snow (0.42 m⁻¹ as per Davis, 1996), and $z$ (m) is the depth in the snow layer. The fractions of penetrating solar radiation determined from Eqs. (18) and (21) are used to update the ice and snow temperature profiles as computed from Eqs. (1) and (7), respectively. In Eqs. (1) and (7), the quantity $F_c$ is replaced with $F_{i0}$ and $F_{i0,s}$, respectively.

### 2.2.3 Meltwater ponds

The model presented here allows for the growth and contraction of meltwater ponds on the surface of the sea ice in terms of the depth and fractional area coverage of the ponds. The rate of meltwater pond depth change (m s⁻¹) is defined by (*e.g.*, EC93 and Holland et al., 2012),

$$\frac{\partial h_p}{\partial t} = \left((1 - A_{pond})\left(\frac{\partial h_i}{\partial t}\right)_0 + \frac{A_{pond}F_p}{L_{fi}} + RF + \frac{SF\rho_s}{\rho_w}\right)(1 - r), \tag{22}$$

where $h_p$ is pond depth (m), $A_{pond}$ is pond fractional area, $\left(\frac{\partial h_i}{\partial t}\right)_0$ is the rate of ice melt at the top surface of the sea ice (m s⁻¹), $F_p$ is the solar radiation flux (W m⁻²) which is absorbed by both the pond water and the underlying sea ice, $L_{fi}$ is the latent heat of fusion of sea ice (3.014×10⁸ J m⁻³), $RF$ and $SF$ are rainfall and snowfall, respectively (m s⁻¹), $\rho_w$ is the density of melt-pond water (1000 kg m⁻³), and $r$ is the fraction of surface meltwater and rainfall which runs off the ice surface while the rest remains in the ponds. The quantity $r$ is defined as $0.85 - 0.7A_{ice}$ (Holland et al., 2012), where $A_{ice}$ is the fractional ice area concentration of a given model particle. As per EC93, once the melt-pond depth on a given model particle is greater than zero, the melt-pond fractional area is initialized at 0.10.

The flux of solar radiation absorbed by the pond water and sea ice is defined by (*e.g.*, EC93),

$$F_p = F_{SW}\left(a_p + a_p\alpha_i t_p + t_p(1 - \alpha_i)(1 - I_0)\right)(1 - \alpha_p), \tag{23}$$

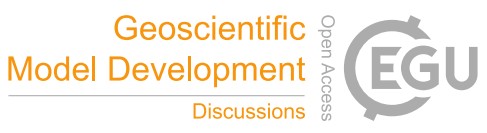

where $a_p$ is the pond absorption coefficient ($1 - t_p^{0.89}$), $t_p$ is the pond transmissivity ($0.36 - 0.17 \log_{10}(h_p)$), and
$\alpha_p$ is the pond albedo. As per EC93, the maximum allowed melt-pond depth is set to 0.8 m. Any melt-ponds which
grow to a depth greater than the ice thickness (when the ice thickness is less than 0.8 m) are subsequently treated as
leads in the ice pack (see subsection 2.2.4).
One melt ponds have begun to form, the melt-pond fractional area is computed iteratively using the scheme of
Holland et al., 2012 which allows for growth and contraction of the ponds based on the ice surface temperature. The
evolution of the melt-pond fractional area in terms of growth and contraction is defined respectively as follows,
$$A_{pond(i+1)} = A_{pond(i)} + \frac{(1-r)\left(\left(\frac{\partial h_i}{\partial t}\right)_0 \left(\frac{\rho_i}{\rho_w}\right) + RF\right)}{h_p} dt \qquad\qquad \text{for } T_{i,s} > \text{-2°C},$$
$$A_{pond(i+1)} = A_{pond(i)} e^{\frac{0.01\left(T_{i,s}+2\right)}{2}} \qquad\qquad\qquad \text{for } T_{i,s} \leq \text{-2°C}, \qquad (24)$$
where $i$ is the model time-step index, and $dt$ is the model time-step (seconds). As per EC93, the melt-pond
fractional area is restricted to a maximum value of 0.25. Melt-ponds are not permitted to form until all of the snow
has melted from a model particle.

## 291    2.2.4 Lead development and ocean heat fluxes

The present model uses the parameterizations of EC93 to simulate the ocean-air and ocean-ice heat exchanges and
the development of leads in the ice pack. The net heat flux (W m⁻²) at the lead surface is defined by (*e.g.*, EC93),
$$F_{lead} = \epsilon_w F_{LW} - \epsilon_w \sigma T_w^4 + (1 - \alpha_w)(1 - i_w)F_{SW} - F_{sens,w} - F_{lat,w}, \qquad (25)$$
where $\epsilon_w$ is the emissivity of water (0.97), $F_{LW}$ is the incoming (downward) infrared heat flux from the atmosphere
(W m⁻²), $\sigma$ is the Stefan-Boltzmann constant (5.67×10⁻⁸ W m⁻² K⁻⁴), $T_w$ is the surface ocean temperature (K), $\alpha_w$ is
the surface ocean albedo, $i_w$ is the fraction of solar radiation transmitted through the surface water layer equivalent
in depth to the ice thickness and not absorbed, and $F_{sens,w}$ and $F_{lat,w}$ are the sensible and latent heat fluxes from the
ocean surface layer, respectively (W m⁻²). The parameter $i_w$ is defined as follows (*e.g.*, EC93),
$$i_w = 1 - \frac{(a_1 + a_2 \ln(h_i))}{1 - \alpha_w}, \qquad (26)$$
where $a_1 = 0.5676$ and $a_2 = 0.1046$ under clear-sky conditions and $a_1 = 0.3938$ and $a_2 = 0.1208$ under cloudy-
sky conditions. In the present model, conditions are considered to be clear-sky if the total cloud area fraction is less
than 0.35.
The growth of the fractional lead area is defined by (*e.g.*, EC93),
$$\frac{\partial A_{lead}}{\partial t} = \frac{A_{lead} F_{lead}}{L_{fi} h_i + L_{fs} h_s}, \qquad (27)$$
where $L_{fs}$ is the latent heat of fusion of snow (1.097×10⁸ J m⁻³), and $h_s$ is the thickness of the snow layer on the ice
(m).





The expression for the solar radiation flux (Wm$^{-2}$) absorbed beneath the ice is represented as the sum of the solar
flux through the leads and the solar radiation transmitted through snow-free ice and is given as follows (*e.g.*, EC93),
$$F_{wi} = A_{lead}(1 - \alpha_w)i_w F_{SW} + (1 - A_{lead})(1 - \alpha_i)I_0 e^{-\kappa_i h_i}F_{SW}. \qquad (28)$$
The water temperature immediately beneath the ice, $T_{wi}$, is governed by (*e.g.*, EC93),
$$\frac{\partial T_{wi}}{\partial t} = \frac{F_{wi} - (1 - A_{lead})F_b}{\left(d_w - \left(\frac{\rho_i}{\rho_w}\right)h_i\right)(\rho c)_w}, \qquad (29)$$
where $F_b$ is the heat flux at the base of the ice (W m$^{-2}$), $d_w$ is the ocean mixed layer depth (m), and $(\rho c)_w$ is the
volumetric heat capacity of water ($4.19 \times 10^6$ J m$^{-3}$ K$^{-1}$). In deeper water, the ocean mixed layer depth is assumed to
be 30 m (EC93), and is assumed to be the bathymetric water depth in shallower water less than 30 m deep (see
Figure 5). The model is initialized with $T_{wi} = T_{o,f}$ as per Eq. (6). The heat flux at the base of the ice is given by
(*e.g.*, EC93),
$$F_b = (\rho c)_w C_{Tb}(T_{wi} - T_{o,f}), \qquad (30)$$
where $C_{Tb}$ is the bulk transfer coefficient. As per EC93, $C_{Tb}$ is defined as,
$$C_{Tb} = 1.26 \times 10^{-4} h_i^{-0.5} \qquad\qquad\qquad \text{for } h_i < 3 \text{ m,}$$
$$C_{Tb} = 7.27 \times 10^{-5} \qquad\qquad\qquad\qquad \text{for } h_i \geq 3 \text{ m.} \qquad (31)$$
**2.2.5 Atmospheric properties and heat fluxes**
The ocean and ice atmospheric heat fluxes in the present model are defined by the bulk aerodynamic formulae for
sensible, latent, and longwave radiative heat transfer as per EC93. The sensible heat flux over ice (W m$^{-2}$) is defined
as (*e.g.*, EC93),
$$F_{sens,i} = \rho_a c_{pa} C_{Ti} U_a (T_{i,s} - T_a), \qquad (32)$$
where $\rho_a$ is the air density (kg m$^{-3}$) at 2 m, $c_{pa}$ is the specific heat of air at constant pressure (J kg$^{-1}$ K$^{-1}$), $C_{Ti}$ is the
bulk heat and moisture transfer coefficient over ice, and $U_a$ is the wind speed at 2 m (m s$^{-1}$). The air density is
computed according to the ideal gas law as (*e.g.*, Tsonis, 2002),
$$\rho_a = \frac{P_s}{R T_a}, \qquad (33)$$
where $P_s$ is the surface or sea level atmospheric pressure (Pa), $R$ is the gas constant for air (J kg$^{-1}$ K$^{-1}$), and the 2 m
air temperature $T_a$ is in K. The gas constant for air depends on the air specific humidity as follows (*e.g.*, Tsonis,

333    2002),

$$R = R_d(1 + 0.61 q_a), \qquad (34)$$
where $R_d$ is the gas constant for dry air (278.058 J kg$^{-1}$ K$^{-1}$), and $q_a$ is the specific humidity of the air at 2 m. The air
specific humidity is defined by (*e.g.*, Tsonis, 2002),



$$q_a = \frac{\frac{0.622 e_w}{P_s - e_w}}{1 + \left(\frac{0.622 e_w}{P_s - e_w}\right)},$$ (35)

where $e_w$ is the vapor pressure in the air (Pa) at 2 m. The vapor pressure is equivalent to $(RH)(e_{sw})$, where $RH$ is the relative humidity at 2 m, and $e_{sw}$ is the saturation vapor pressure (Pa) at 2 m. The saturation vapor pressure is given by (*e.g.*, Tsonis, 2002),

$$e_{sw} = 611 e^{19.83 - \frac{5417}{T_a}},$$ (36)

where $T_a$ is in K. The specific heat of air at constant pressure is defined as $\frac{7}{2} R$ (Tsonis, 2002). The bulk heat and moisture transfer coefficient over ice is defined as (*e.g.*, EC93),

$$C_{Ti} = C_{T0i} \left(1 - \frac{2 b_a Ri_{ice}}{1 + c|Ri_{ice}|^{0.5}}\right) \qquad \text{for } Ri_{ice} < 0,$$

$$C_{Ti} = C_{T0i} (1 + b_a Ri_{ice})^{-2} \qquad \text{for } Ri_{ice} \geq 0,$$ (37)

where $C_{T0i}$ is a heat and moisture transfer coefficient for a neutral surface layer over ice ($1.3 \times 10^{-3}$), $b_a$ is a parameter for atmospheric turbulence equal to 20, $Ri_{ice}$ is the bulk Richardson number over ice, and $c$ has a value of $(1961 b_a C_{T0i})$ for an ice roughness length of $1.6 \times 10^{-4}$ m. The Richardson number is defined as (*e.g.*, EC93),

$$Ri_{ice} = \frac{g(T_a - T_{i,s})\Delta z}{T_a U_a^2},$$ (38)

where $g$ is the earth gravitational acceleration (9.81 m s$^{-2}$), and $\Delta z$ is the height of the wind speed and air temperature data (2 m).

The latent heat flux over sea ice (W m$^{-2}$) is given as (*e.g.*, EC93),

$$F_{lat,i} = \rho_a L_v C_{Ti} U_a (q_{sat(Ti,s)} - q_a),$$ (39)

where $L_v$ is the latent heat of vaporization ($2.501 \times 10^6$ J m$^{-3}$), and $q_{sat(Ti,s)}$ is the saturation specific humidity over ice. This quantity is equal to (*e.g.*, WHOI, 2010),

$$q_{sat(Ti,s)} = \frac{0.622 e_{sat,i}}{P_s - 0.378 e_{sat,i}},$$ (40)

where $e_{sat,i}$ is the saturation vapor pressure directly over the ice (Pa), which is equivalent to (*e.g.*, WHOI, 2010),

$$e_{sat,i} = 10^{\frac{0.7859 + 0.03477 T_{i,s}}{1 + 0.00412 T_{i,s}} + 0.00422 T_{i,s} + 2}.$$ (41)

As per EC93, the formulas for the sensible and latent heat fluxes over leads and open water (W m$^{-2}$), $F_{sens,w}$ and $F_{lat,w}$, respectively, are similar to those for ice (Eqs. 32 and 39). For the sensible heat flux over water, the quantity $T_{i,s}$ in Eq. (32) is replaced with $T_w$ to represent the surface water temperature. In Eq. (32), the bulk heat and moisture transfer coefficient over water, $C_{Tw}$, is determined using Equation 37 with $C_{T0} = 1.0 \times 10^{-3}$ over water, and the bulk Richardson number over water, $Ri_{water}$, is calculated by replacing $T_a$ with $T_w$ in Eq. (38). The latent heat flux over water is computed by replacing $C_{Ti}$ with $C_{Tw}$ in Equation 39, and $q_{sat(Ti,s)}$ with $q_{sat(Tw)}$, the saturation specific





humidity over water. The quantity is computed from Equation 40, however saturation vapor pressure directly over
water, $e_{sat,w}$ (Pa), is determined by replacing $T_{i,s}$ with $T_w$ in Equation 41, and multiplying Eq. (41) by 0.98.
The downward longwave (infrared) radiative flux from the atmosphere to the ocean and ice (Wm$^{-2}$) is defined by
(*e.g.*, Konig-Langlo and Augstein, 1994),
$$F_{LW} = (0.765 + 0.22T_{cc}^3)\sigma T_a^4, \tag{42}$$
where the quantity $(0.765 + 0.22T_{cc}^3)$ represents the atmospheric emissivity as a function of the total fractional
cloud cover area.
**2.2.6 Albedo**
The surface albedo parameterization follows the scheme used in EC93, in which there are five surface types
considered: dry snow, melting snow, bare ice, meltwater ponds, and open water. The albedo scheme additionally
considers the spectral variation in the incoming shortwave radiation in four ranges of wavelength, as well as the
albedo dependence on solar zenith angle. The solar zenith angle is given by,
$$\theta_0 = \mathrm{acos}(\mu_0), \tag{43}$$
where the quantity $\mu_0$ is defined as,
$$\mu_0 = \sin(\phi)\sin(\delta) + \cos(\phi)\cos(\delta)\cos(\theta_h). \tag{44}$$
In Eq. (44), $\delta$ is defined as (*e.g.*, NOAA, 2015),
$$\delta = 0.006918 - 0.399912\cos(d) + 0.070257\sin(d) - 0.006758\cos(2d) + 0.000907\sin(2d) -$$
$$0.002697\cos(3d) + 0.00148\sin(3d), \tag{45}$$
where the quantity $d$ is defined by $d = \frac{2\pi(J-1)}{365.2422}$ and is given in radians. In Eq. (44), $\theta_h$ is the hour angle, and is
defined as $\theta_h = 15(T_{hr} - M) - L_m$, in which $T_{hr}$ is the hour of the day (*e.g.*, 13 for 13:00) and $M$ is defined by
(*e.g.*, NCEP, 2015),
$$M = 12 + 0.12357\sin(d) - 0.004289\cos(d) + 0.153809\sin(2d) + 0.060783\cos(2d). \tag{46}$$
The relative weights assigned to each wavelength band of the solar spectrum, as well as the fractions of diffuse
versus direct radiation, are summarized in Table 2 as a function of month (*e.g.*, EC93).
**Table 2.** Summary of solar spectrum relative weights as a function of wavelength band and month, as well as
fractions of diffuse versus direct radiation (EC93).

| Solar Spectrum Weights and Radiation Fractions for Surface Albedo | | | | | | |
|---|---|---|---|---|---|---|
| **Month** | **Wavelength Band (µm)** | | | | **Diff. Rad. (DR) Fraction** | **Direct Rad. (1 – DR) Fraction** |
| | **0.25 - 0.69** | **0.69 - 1.19** | **1.19 - 2.38** | **2.38 - 4.00** | | |
| **Jan** | 0.520 | 0.343 | 0.129 | 0.008 | 0.779 | 0.221 |




| | | | | | |
|---|---|---|---|---|---|
| **Feb** | 0.520 | 0.343 | 0.129 | 0.008 | 0.779 | 0.221 |
| **Mar** | 0.503 | 0.343 | 0.142 | 0.012 | 0.658 | 0.342 |
| **Apr** | 0.492 | 0.339 | 0.153 | 0.016 | 0.489 | 0.511 |
| **May** | 0.504 | 0.338 | 0.144 | 0.014 | 0.581 | 0.419 |
| **Jun** | 0.527 | 0.340 | 0.124 | 0.009 | 0.724 | 0.276 |
| **Jul** | 0.545 | 0.315 | 0.130 | 0.010 | 0.698 | 0.302 |
| **Aug** | 0.539 | 0.321 | 0.130 | 0.010 | 0.715 | 0.285 |
| **Sep** | 0.517 | 0.339 | 0.134 | 0.010 | 0.717 | 0.283 |
| **Oct** | 0.519 | 0.343 | 0.130 | 0.008 | 0.790 | 0.210 |
| **Nov** | 0.520 | 0.343 | 0.129 | 0.008 | 0.779 | 0.221 |
| **Dec** | 0.520 | 0.343 | 0.129 | 0.008 | 0.779 | 0.221 |

The albedo is then given as (*e.g.*, EC93),
$$\alpha = (DR)\left(\Sigma_{j=1}^{4} w_j \alpha_{j(DR)}\right) + (1 - DR)\left(\Sigma_{j=1}^{4} w_j \alpha_{j(1-DR)}\right),$$ (47)
where $w_j$ is the weight of a given wavelength band, $\alpha_j$ is the albedo in that band, $DR$ refers to the fraction of diffuse
radiation, and $(1 - DR)$ refers to the fraction of direct radiation. Table 3 summarizes the albedo values as functions
of the wavelength band of the incoming shortwave radiation spectrum, the surface type, direct or diffuse radiation,
and ice and snow thickness, and is reproduced here from Table 2 in EC93.
**Table 3.** Spectral albedos for the various surface types, reproduced here from Table 2 in EC93.

| **Spectral Albedos for Five Surface Types** | | | | |
|---|---|---|---|---|
| **Surface Type** | **Wavelength Band (μm)** | | | |
| | **0.25 - 0.69** | **0.69 - 1.19** | **1.19 - 2.38** | **2.38 - 4.00** |
| **Dry snow, $\alpha_s$** | | | | |
| **Direct Rad.** | $0.980 - 0.008\mu_0$ | $0.902 - 0.116\mu_0$ | $0.384 - 0.222\mu_0$ | $0.053 - 0.047\mu_0$ |
| **Diffuse Rad.** | 0.975 | 0.832 | 0.250 | 0.025 |
| **Melting snow, $\alpha_s$** | | | | |
| **$h_s \geq 0.1m$** | 0.871 | 0.702 | 0.079 | 0.010 |
| **$h_s < 0.1m$** | linearly reduced to bare sea ice value | linearly reduced to bare sea ice value | linearly reduced to bare sea ice value | linearly reduced to bare sea ice |





| | | | value |
|---|---|---|---|
| **Bare sea ice, $\alpha_i$** | | | |
| **$h_i$ < 1m** | $0.760 + 0.140\ln(h_i)$ | $0.247 + 0.029\ln(h_i)$ | 0.055 | 0.036 |
| **1m ≤ $h_i$ <2m** | $0.770 + 0.018(h_i - 1)$ | $0.247 + 0.196(h_i - 1)$ | 0.055 | 0.036 |
| **$h_i$ ≥ 2m** | 0.778 | 0.443 | 0.055 | 0.036 |
| **Meltwater pond, $\alpha_p$** | $0.150 + \exp(-8.1h_p - 0.47)$ | $0.054 + \exp(-31.8h_p - 0.94)$ | $0.033 + \exp(-2.6h_p - 3.82)$ | 0.030 |
| **Open water/leads, $\alpha_w$** | | | |
| **Direct Rad.** | $\alpha_w^* + 0.008$ | $\alpha_w^* - 0.007$ | $\alpha_w^* - 0.007$ | $\alpha_w^* - 0.007$ |
| **Diffuse Rad.** | 0.060 | 0.060 | 0.060 | 0.060 |

In Table 3, $h_s$ is snow depth (m), and the value $\alpha_w^*$ is defined as (*e.g.*, EC93),
$$\alpha_w^* = \frac{0.026}{\mu_0^{1.7} + 0.065} + 0.015(\mu_0 - 0.1)(\mu_0 - 0.5)(\mu_0 - 1.0). \tag{48}$$
**2.2.7 Energy balance and thickness changes over snow and ice**
The energy balance at the surface of the snow or snow-free ice layer is represented by (*e.g.*, EC93),
$$(F_{net})_0 = \epsilon(F_{LW} - \sigma T_0^4) + (1 - \alpha)(1 - I_0)F_{SW} - F_{sens} - F_{lat} = -(F_c)_0, \tag{49}$$
where $\epsilon$ is the emissivity of the snow or ice (0.99), $T_0$ is the temperature of the snow or ice surface (K), $\alpha$ is the
snow or ice albedo, and $(F_c)_0$ is the conductive heat flux at the snow or ice surface (W m$^{-2}$). Eq. (49) is used to solve
for the snow or snow-free ice surface temperature after the first time-step.
Changes in snow and ice thickness are governed by the energy balances at the snow and ice surfaces and the base of
the ice, and the volumetric heats of fusion for snow and ice. The rate of change in snow thickness at the surface is
given by (*e.g.*, EC93),
$$\left(\frac{\partial h_s}{\partial t}\right)_0 = \frac{-F_{net} - (F_c)_0}{L_{fs}}, \tag{50}$$
where $L_{fs}$ is the volumetric heat of fusion for snow (1.097×10$^8$ J m$^{-3}$). Similarly, the rate of change in thickness at
the surface of the snow-free ice is given by (*e.g.*, EC93),
$$\left(\frac{\partial h_i}{\partial t}\right)_0 = \frac{-F_{net} - (F_c)_0}{L_{fi}}, \tag{51}$$
where $L_{fi}$ is the volumetric heat of fusion for ice (3.014×10$^8$ J m$^{-3}$). Changes in thickness at the ice surface are
constrained to be $\left(\frac{\partial h_i}{\partial t}\right)_0 \leq 0$ so that ice growth only occurs through accretion at the base. The accretion or melting
at the base of the ice is defined by (*e.g.*, EC93),





$$\left(\frac{\partial h_i}{\partial t}\right)_b = \frac{(F_c)_b - F_b}{L_{fb}},$$ (52)
where $(F_c)_b$ is the conductive heat flux at the base of the ice (W m$^{-2}$), and $L_{fb}$ is the volumetric heat of fusion for ice
at the base ($2.679 \times 10^8$ J m$^{-3}$).
**2.3 Dynamic model description**
The dynamic model is based on the Smoothed Particle Hydrodynamics (SPH) formulation as originally applied to
ice dynamics in Gutfraind and Savage, 1997 and Lindsay and Stern, 2004. The dynamic model is coupled to the
thermodynamic model, and is run at each time-step immediately after the thermodynamic model. It is forced by
surface wind speed and direction, surface current speed and direction, Coriolis deflection, the horizontal gradient of
the sea surface slope, internal ice stresses, and a coastal boundary force. As per Lindsay and Stern, 2004, particles
which drift within 0.4 of the initial grid spacing are combined into a single particle.
**2.3.1 Air drag**
The zonal and meridional components of the acceleration (m s$^{-2}$) on each Lagrangian ice particle due to wind forcing
are given by,
$$\overrightarrow{A_{wx}} = \left(\frac{1}{\rho_i h_i}\right)\rho_a C_a \sqrt{(\overrightarrow{u_a} - \overrightarrow{u_i})^2 + (\overrightarrow{v_a} - \overrightarrow{v_i})^2}\,(\overrightarrow{u_a} - \overrightarrow{u_i}),$$
$$\overrightarrow{A_{wy}} = \left(\frac{1}{\rho_i h_i}\right)\rho_a C_a \sqrt{(\overrightarrow{u_a} - \overrightarrow{u_i})^2 + (\overrightarrow{v_a} - \overrightarrow{v_i})^2}\,(\overrightarrow{v_a} - \overrightarrow{v_i}),$$ (53)
respectively, where $C_a$ is the dimensionless air drag coefficient ($2.0 \times 10^{-3}$), $\overrightarrow{u_a}$ and $\overrightarrow{u_i}$ are the zonal velocities for
wind and ice drift, respectively, and $\overrightarrow{v_a}$ and $\overrightarrow{v_i}$ are the meridional velocities for wind and ice drift, respectively (m s$^{-1}$

433 ).

**2.3.2 Water drag**
The zonal and meridional components of the acceleration (m s$^{-2}$) on the ice particles due to ocean current forcing are
similar to those for the wind and are given by,
$$\overrightarrow{A_{ox}} = \left(\frac{1}{\rho_i h_i}\right)\rho_{wo} C_w \sqrt{(\overrightarrow{u_w} - \overrightarrow{u_i})^2 + (\overrightarrow{v_w} - \overrightarrow{v_i})^2}\,(\overrightarrow{u_w} - \overrightarrow{u_i}),$$
$$\overrightarrow{A_{oy}} = \left(\frac{1}{\rho_i h_i}\right)\rho_{wo} C_w \sqrt{(\overrightarrow{u_w} - \overrightarrow{u_i})^2 + (\overrightarrow{v_w} - \overrightarrow{v_i})^2}\,(\overrightarrow{v_w} - \overrightarrow{v_i}),$$ (54)
respectively, where $\rho_{wo}$ is the density of seawater (kg m$^{-3}$), $C_w$ is the dimensionless water drag coefficient ($5.0 \times 10^{-3}$
), and $\overrightarrow{u_w}$ and $\overrightarrow{v_w}$ are the zonal and meridional velocities for the surface current, respectively (m s$^{-1}$). The density of
seawater is computed by (*e.g.*, Fofonoff and Millard Jr., 1983),
$\rho_{wo} = 999.842594 + 6.793952 \times 10^{-2} T_w - 9.09529 \times 10^{-3} T_w^2 + 1.001685 \times 10^{-4} T_w^3 - 1.120083 \times$
$10^{-6} T_w^4 + 6.536332 \times 10^{-9} T_w^5 + (8.24493 \times 10^{-1} - 4.0899 \times 10^{-3} T_w + 7.6438 \times 10^{-5} T_w^2 - 8.2467 \times$





$10^{-7}T_w^3 + 5.3875 \times 10^{-9}T_w^4)S + (-5.72466 \times 10^{-3} + 1.0227 \times 10^{-4}T_w - 1.6546 \times 10^{-6}T_w^2)S^{1.5} + 4.8314 \times$
$10^{-4}S^2,$ (55)
where $T_w$ is the SST (°C).
**2.3.3 Coriolis deflection**
The zonal and meridional components of the acceleration (m s$^{-2}$) on the ice particles due to Coriolis forcing are
given by,
$\overrightarrow{A_{cx}} = f\overrightarrow{v_i},$
$\overrightarrow{A_{cy}} = -f\overrightarrow{u_i},$ (56)
respectively. In Eq. (56), $f$ is the Coriolis parameter (s$^{-1}$) and is defined by $2\omega_e\sin(\phi)$, where $\omega_e$ is the Earth
angular speed (7.292115×10$^{-5}$ rad s$^{-1}$).
**2.3.4. Sea surface gradient forcing**
The zonal and meridional components of the horizontal gravitational acceleration (m s$^{-2}$) on the ice particles due to
the sea surface gradient are given by,
$\overrightarrow{A_{gx}} = -g\overrightarrow{\nabla H_x},$
$\overrightarrow{A_{gy}} = -g\overrightarrow{\nabla H_y},$ (57)
respectively, where $\overrightarrow{\nabla H_x}$ and $\overrightarrow{\nabla H_y}$ are the zonal and meridional gradients of the sea surface.
**2.3.5. Internal ice stresses**
The first step in calculating the internal ice stresses is to project the ice particle positions from their latitude-
longitude grid onto a Cartesian grid in order to facilitate computations of distance between the particles. The
northing-easting grid chosen for this work is the Mercator projection. Conversions of latitude and longitude to
Mercator northing, y, and easting, x, coordinates (m) are defined by,
$y = R_e k ln\left(\tan\left(\frac{\pi}{4} + \frac{\phi}{2}\right)\right),$
$x = R_e kL,$ (58)
respectively, where $R_e$ is the radius of the earth (6371×10$^3$ m), $k$ is the scale factor for 55°N latitude (1.75), and $L$ is
the longitude (radians).
The zonal and meridional components of the acceleration (m s$^{-2}$) on the ice particles due to internal stress gradients
are given by (*e.g.*, Lindsay and Stern, 2004),
$\overrightarrow{A_{sx}} = \left(\frac{1}{\rho_i h_i}\right)\left(\frac{\partial\sigma_{xx}}{\partial x} + \frac{\partial\sigma_{xy}}{\partial y}\right),$




$\quad \overrightarrow{A_{sy}} = \left(\frac{1}{\rho_i h_i}\right)\left(\frac{\partial \sigma_{yy}}{\partial y} + \frac{\partial \sigma_{xy}}{\partial x}\right),$ (59)
$\quad$ respectively, where $\sigma$ is the internal ice stress tensor (kg s$^{-2}$). The components of the stress gradients $\frac{\partial \sigma_{xx}}{\partial x}, \frac{\partial \sigma_{yy}}{\partial y}, \frac{\partial \sigma_{xy}}{\partial x}$,
$\quad$ and $\frac{\partial \sigma_{xy}}{\partial y}$ are computed by first calculating the components of the stress tensor at the ice particle positions and then
$\quad$ linearly interpolating these values to four points surrounding each ice particle (Figure 6).

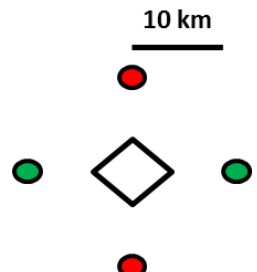


$\quad$ **Figure 6.** Schematic of the locations at which the ice stress tensor components are computed for calculation of the
$\quad$ stress gradients. The white diamond represents an ice particle, the green circles represent the locations between
$\quad$ which the zonal stress gradients are computed, and the red circles represent the locations between which the
$\quad$ meridional stress gradients are computed. The stress gradient length scale is 10 km (image is slightly altered from
$\quad$ original version in Lindsay and Stern, 2004).
$\quad$ In Figure 6, an ice particle is represented by the white diamond. The zonal components of the stress gradients, $\frac{\partial \sigma_{xx}}{\partial x}$
$\quad$ and $\frac{\partial \sigma_{xy}}{\partial x}$, are computed by finite differencing of the stress tensor values at the green circles, and the meridional
$\quad$ components of the stress gradients, $\frac{\partial \sigma_{yy}}{\partial y}$ and $\frac{\partial \sigma_{xy}}{\partial y}$, are computed by finite differencing of the stress tensor values at
$\quad$ the red circles.
$\quad$ The values of the stress tensor (kg m s$^{-2}$) are calculated as follows for each ice particle (*e.g.*, Gutfraind and Savage,
$\quad$ 1997),
$\quad$ $\sigma_{xx} = P_i - 2\eta\left(\dot{\epsilon}_{xx} - 0.5\left(\dot{\epsilon}_{xx} + \dot{\epsilon}_{yy}\right)\right),$
$\quad$ $\sigma_{yy} = \frac{\sigma_{xx}(1-\sin(\phi_i))}{1+\sin(\phi_i)},$
$\quad$ $\sigma_{xy} = -2\eta\dot{\epsilon}_{xy},$ (60)
$\quad$ where $P_i$ is the internal ice pressure (N m$^{-2}$), $\eta$ is the ice viscosity (kg m$^{-1}$ s$^{-1}$), $\dot{\epsilon}_{xx}$, $\dot{\epsilon}_{yy}$, and $\dot{\epsilon}_{xy}$ are the components
$\quad$ of the strain rate tensor (s$^{-1}$), and $\phi_i$ is the internal ice friction angle (approximately 17.5°). The evaluation of $\sigma_{yy}$ as
$\quad$ a function of the other principal stress component, $\sigma_{xx}$, satisfies the Mohr-Coulomb yield criterion for a plastic
$\quad$ regime as per Gutfraind and Savage, 1997. In Eq. (60), the pressure is taken as the ice compressive strength as per
$\quad$ Hibler III, 1979 and Sayed et al., 2002 and is defined as,



$P_i = P^* h_i e^{-20(1-A_{ice})}$,  (61)
where $P^*$ is a sea ice strength constant ($5.0 \times 10^3$ N m$^{-2}$). The ice viscosity is defined as (*e.g.*, Gutfraind and Savage,

498     1997),

$\eta = \min\left(\dfrac{P_i \sin(\phi_i)}{\dot{\epsilon}_{xx} - \dot{\epsilon}_{yy}}, \eta_{max}\right)$,  (62)
where $\eta_{max}$ is the maximum possible value of the ice viscosity ($1.0 \times 10^{11}$ kg m$^{-1}$ s$^{-1}$).
The components of the strain rate tensor are evaluated using the SPH formulation. While the strain rates for any
given ice particle are theoretically influenced by all other ice particles in the model domain, it is most
computationally practical to assume only a limited number of nearest-neighbor ice particles influence the strain rates
of a given particle. In this paper, only the nearest three ice particles are considered when calculating strain rates for a
given particle. Coastal boundary particles are considered when finding the nearest-neighbor particles, and their
influence on the strain rates of nearby ice particles is treated the same as that from other ice particles. However,
coastal cells have fixed zero-velocities and are assumed to be constantly circular with a radius of 5 km. They also
exert an additional repulsive force on ice particles (see section 2.3.6).
The influence that a given ice particle exerts on another decreases exponentially with distance between particles, and
is most commonly represented through the Gaussian kernel. The two-dimensional Gaussian kernel is defined as
(*e.g.*, Gutfraind and Savage, 1997),
$W(\boldsymbol{r}, L) = \dfrac{1}{\pi L^2} e^{-\frac{r^2}{L^2}}$,  (63)
where $\boldsymbol{r}$ is the distance from the ice particle for which the strain rates are being calculated to a nearest-neighbor
particle (m), and $L$ is a smoothing length (150 km as per Lindsay and Stern, 2004). The components of the strain rate
tensor are then given as (*e.g.*, Gutfraind and Savage, 1997),
$\dot{\epsilon}_{xx} = \Sigma_k \left( -\dfrac{2 I_{area_k} (u_k - u_i) W(\boldsymbol{r}, L)_k r_{x_k}}{L^2} \right)$,
$\dot{\epsilon}_{yy} = \Sigma_k \left( -\dfrac{2 I_{area_k} (v_k - v_i) W(\boldsymbol{r}, L)_k r_{y_k}}{L^2} \right)$,
$\dot{\epsilon}_{xy} = \Sigma_k \left( \dfrac{-I_{area_k} (v_k - v_i) W(\boldsymbol{r}, L)_k r_{y_k} - I_{area_k} (u_k - u_i) W(\boldsymbol{r}, L)_k r_{x_k}}{L^2} \right)$,  (64)
where $I_{area_k}$ is the total surface are of the $k^{th}$ nearest-neighbor ice particle (m$^2$), $u_k$ and $v_k$ are the zonal and
meridional velocities of the $k^{th}$ nearest-neighbor ice particles, respectively (m s$^{-1}$), $u_i$ and $v_i$ are the zonal and
meridional velocities of the ice particle for which the strain rate tensor is being calculated, respectively (m s$^{-1}$), and
$r_{x_k}$ and $r_{y_k}$ are the zonal and meridional distances from the ice particle for which the strain rates are being
calculated to the $k^{th}$ nearest-neighbor particle (m). The surface areas of the ice particles are calculated at each time-
step using Voronoi tessellation over the whole model domain as per Lindsay and Stern, 2004. For each ice particle,
the Voronoi polygon covers the area surrounding the particle which is closer to that particle than to any other.





**2.3.6 Coastal boundary force**
In order to prevent ice particles from drifting past the coastal boundary cells, a repulsive force from each coastal cell
is exerted on every ice particle in the direction of the vector pointing from the coastal cell toward the ice particle. As
per Lindsay and Stern, 2004, the force is given in the form $\frac{1}{R^2}$, where $\boldsymbol{R}$ is the distance from the coastal cell to the ice
particle. The force is set to be 0.1 N m$^{-2}$ at a distance of 25 km. An identical repulsion force is also exerted on ice
particles from the boundaries of the model domain in the ocean in order to keep ice particles from drifting outside
the model domain.
**2.3.7 Dynamic evolution of ice thickness and concentration**
Changes in the ice thickness and concentration of particles due to dynamics are determined by the principal
components of the strain rate tensor as follows,
$$\frac{\partial h_i}{\partial t} = -h_i\big(\dot{\epsilon}_{xx} + \dot{\epsilon}_{yy}\big),$$
$$\frac{\partial A_{ice}}{\partial t} = -A_{ice}\big(\dot{\epsilon}_{xx} + \dot{\epsilon}_{yy}\big), \qquad (65)$$
respectively. The dynamically induced changes in ice thickness and concentration are then added to any
thermodynamically induced changes in these parameters at each time-step.
**2.4 Time-stepping schemes**
Several numerical integration schemes for time-stepping are used in the present model. The Euler-forward scheme
(*e.g.*, Khandekar, 1980) is used to solve the thermodynamic equations for changes in snow and ice thickness (Eqs.
50-52), changes in snow and ice temperature due to solar radiation penetration (Eqs. 18, 20-21), and snow
accumulation. This scheme is used to calculate changes from the first to second time-step in melt-pond depth (Eq.
22), water temperature beneath the base of the ice (Eq. 29), and lead area fraction (Eq. 27). The Euler-forward
scheme is additionally used in the dynamic model to integrate the ice particle velocities to position displacements, to
update the dynamically forced changes in ice thicknesses and concentrations (Eq. 65), and to integrate the initial ice
accelerations to velocities from the first time-step to the second. The Euler-forward scheme is represented by,
$$X_{i+1} = X_i + \Delta t \left(\frac{\partial F}{\partial t_i}\right) : i = 1, \qquad (66)$$
where $X$ is the variable being solved, $i$ is the time-step index, $\Delta t$ is the time-step (seconds), and $F$ is the function of
$X$ differentiated with respect to time, $t$.
The second-order Adams-Bashforth scheme (*e.g.*, Khandekar, 1980) is used to solve thermodynamic equations for
changes in melt-pond depth (Eq. 22), water temperature beneath the base of the ice (Eq. 29), and lead area fraction
(Eq. 27) from the second to the fourth time-steps. This scheme is also used to solve the changes in ice particle
velocities from the second to the fourth time-steps in the dynamic model. The second-order Adams-Bashforth
scheme is represented by,



$\qquad X_{i+1} = X_i + \Delta t \left(1.5 \frac{\partial F}{\partial t_i} - 0.5 \frac{\partial F}{\partial t_{i-1}}\right) : 1 < i \leq 3.$ (67)
The fourth-order Adams-Bashforth scheme (*e.g.*, Matthews and Fink, 2004) is used to solve for changes in melt-
pond depth (Eq. 22), water temperature beneath the base of the ice (Eq. 29), and lead area fraction (Eq. 27) from the
fourth time-step onward in the thermodynamic model. In the dynamic model, the fourth-order Adams-Bashforth
predictor-corrector scheme (*e.g.*, Matthews and Fink, 2004) is used to solve the ice velocity changes from the fourth
time-step onward. The fourth-order Adams-Bashforth predictor-corrector scheme is represented by,
$\qquad X_{i+1} = X_i + \frac{\Delta t}{24}\left(55 \frac{\partial F}{\partial t_i} - 59 \frac{\partial F}{\partial t_{i-1}} + 37 \frac{\partial F}{\partial t_{i-2}} - 9 \frac{\partial F}{\partial t_{i-3}}\right),$
$\qquad X_{i+1} = X_i + \frac{\Delta t}{24}\left(9 \frac{\partial F}{\partial t_{i+1}} + 19 \frac{\partial F}{\partial t_i} - 5 \frac{\partial F}{\partial t_{i-1}} + \frac{\partial F}{\partial t_{i-2}}\right) : i > 3,$ (68)
where the first line represents the fourth-order Adams-Bashforth scheme, and both lines represent the sequential
steps of the combined predictor-corrector scheme.
The one-dimensional heat equations for the snow and ice temperatures (Eqs. 1 and 7) in the thermodynamic model
are solved using the standard Euler-forward scheme and centred finite-differencing for second-order differential
equations between the first and second time-steps, and the Dufort-Frankel algorithm (*e.g.*, EC93 and Mitchell and
Griffiths, 1980) from the second time-step onward. The Euler-forward scheme with centred finite-differencing is
represented by (*e.g.*, MacKinnon, 2015),
$\qquad X_{i+1}^j = X_i^j + \frac{\kappa \Delta t}{\Delta z^2}\left(X_i^{j+1} - 2X_i^j + X_i^{j-1}\right) : i = 1,$ (69)
where $j$ is the depth level index in the snow or ice layer, $\kappa$ is the quotient of the thermal conductivity and the
volumetric heat capacity for snow or ice, and $\Delta z$ is the spacing between depth levels (m). The Dufort-Frankel
algorithm is represented by (*e.g.*, MacKinnon, 2015),
$\qquad X_{i+1}^j = \left(\frac{1-\beta}{1+\beta}\right) X_{i-1}^j + \left(\frac{\beta}{1+\beta}\right)\left(X_i^{j+1} + X_i^{j-1}\right),$ (70)
where $\beta$ represents the quantity $2\frac{\kappa \Delta t}{\Delta z^2}$.
Acceptable numerical stability of the integration schemes is achieved when a time-step $\Delta t$ of one hour (3600
seconds) is used for the thermodynamic model-only runs, and a time-step of one minute (60 seconds) is used for the
thermodynamic-dynamic coupled model runs.
**2.5 Model input**
The atmospheric inputs to the model include the 2 m air and dew-point temperatures, relative humidity, 10 m wind
speed and direction, surface pressure, total cloud cover area fraction, snowfall, and total precipitation. All
atmospheric parameters except for precipitation are taken from the North American Regional Reanalysis (NARR),
which has an approximately 0.3° spatial resolution (about 33.3 km) and a three-hourly temporal resolution. In order





to make the wind speed more realistic for near surface winds, a logarithmic wind profile law is used to scale the 10
m height wind speeds to 2 m (*e.g.*, ISO 19901-1),
$U_a = U_{a_{10m}}\left(1 + 0.0573\sqrt{1 + 0.15U_{a_{10m}}}\right)\ln\left(\frac{2}{10}\right),$     (71)
where $U_{a_{10m}}$ is the NARR wind speed at 10 m (m s$^{-1}$).
The snowfall and total precipitation data are given in meters of water equivalent over the dataset time-step and are
taken from the European Centre for Medium-Range Weather Forecasting (ECMWF) ERA-Interim Reanalysis,
which has a 0.5°-0.75° spatial resolution (about 55.5-83 km) and a six-hourly temporal resolution. Rainfall is
assumed to be the difference of the total precipitation and the snowfall. In order to convert the ECMWF snowfall
data from meters of water equivalent to actual snow accumulation, $SF$, the following relationship to 2 m air
temperature is used (*e.g.*, NWS, 1996),
$SF = 5SF_w, T_a \geq 0,$
$SF = 10SF_w, 0 > T_a \geq -1.1,$
$SF = 15SF_w, -1.1 > T_a \geq -3.9,$
$SF = 20SF_w, -3.9 > T_a \geq -7.8,$
$SF = 30SF_w, -7.8 > T_a \geq -11.1,$
$SF = 40SF_w, -11.1 > T_a \geq -15,$
$SF = 50SF_w, T_a < -15,$     (72)
where $SF_w$ is the snowfall rate in meters of water equivalent (m s$^{-1}$), and $T_a$ is in °C.
The ice and oceanographic inputs to the model include the sea surface temperature, sea surface salinity, surface
current speed and direction, sea surface height above the geoid, and sea ice thickness, concentration, and drift speed
and direction. The ice and oceanographic parameters are taken from the Operational Mercator global Ocean analysis
and forecast system. This dataset has a 0.08° spatial resolution (approximately 8.9 km), a two-hourly temporal
resolution for sea surface temperature, sea surface salinity, and surface current speed and direction, and a daily-mean
temporal resolution for sea surface height above the geoid, and sea ice thickness, concentration, and drift speed and
direction. In this dataset, the sea surface temperature, salinity, currents, and height are initially derived from the
Nucleus for European Models of the Ocean (NEMO 3.1), while the sea ice thickness, concentration, and velocity are
derived from the LIM2 sea ice model. Sea surface height anomaly data derived from the Jason2, Cryosat, and Saral-
Altika satellites are assimilated into the dataset, as well as sea surface temperature data from the Reynolds AVHRR-
AMSR ¼° satellite, and in-situ profiles of ocean temperature and salinity. The sea ice thickness, concentration, and
velocity are used to initialize the model, while the remaining metocean parameters described above are used to force
the model for the duration of a run.



Finally, the Labrador coastal boundaries are included, and well as the ocean bathymetry (*e.g.*, GEBCO, 2008, 30-
second data). All dynamic input data are spatially and temporally linearly interpolated as needed.

**2.6 Comparison of LIM2 sea ice data with CIS ice charts**

Given the fact that the LIM2 sea ice dataset used here to initialize the model and compare model results to the "best
estimate" of sea ice conditions is itself derived from a model, it is worth comparing the LIM2 ice data from the
current model run period of April-May 2015 for the Labrador coast with observations from the CIS ice charts. The
LIM2 dataset was used in this study due to its relatively high temporal (daily) resolution compared to the weekly-
mean resolution of the CIS digitized ice charts. Figure 7-Figure 15 show the normalized errors between the weekly
average CIS chart and the LIM2 ice concentration and thickness for April-May 2015 for central coastal Labrador
from Makkovik north to Nain. Here, the normalized (dimensionless) error is defined as,

$$E_{C,T} = \frac{CIS_{C,T} - LIM2_{C,T}}{CIS_{C,T}}, \tag{73}$$

where $CIS_{C,T}$ is the ice concentration (0-1) or thickness (m) of a given domain grid point from the CIS chart, and
$LIM2_{C,T}$ is the ice concentration (0-1) or thickness (m) of the same grid point from the LIM2 dataset.
Figure 7-Figure 15 show that the LIM2 dataset is fairly close to the CIS charts in terms of ice concentration and
thickness for this period in the areas of the model domain closer to shore. The LIM2 dataset simulates greater ice
concentrations and thicknesses further offshore than the CIS charts show. However, the present paper focuses on
simulating ice conditions close to the Labrador coast from near Makkovik up to Nain; hence the LIM2 dataset can
be considered reliable for the purposes of this study. Note that the LIM2 model does not simulate the land-fast ice
cover directly along the shoreline, whereas the CIS charts show the land-fast ice. However, in this paper, the break-
up of the land-fast ice is assumed to occur when the deployed buoys began to drift.

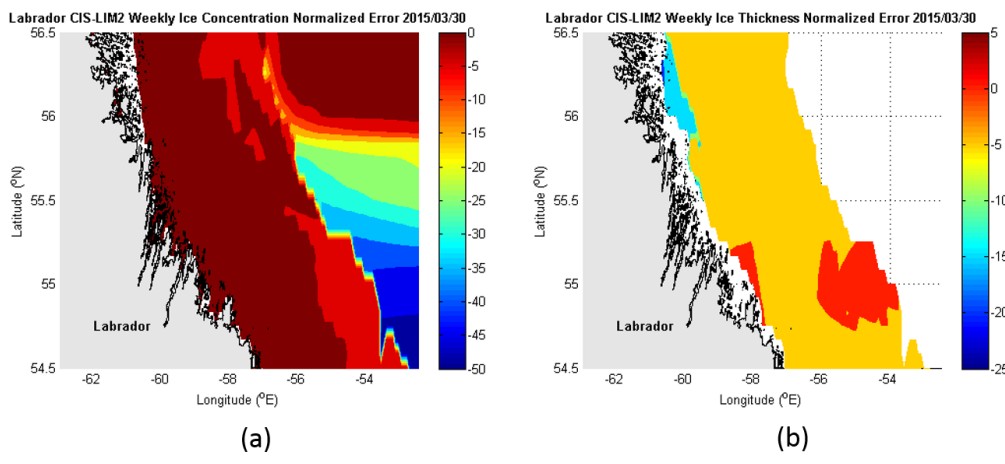


**Figure 7.** Central coastal Labrador weekly average CIS-LIM2 ice concentration (a) and thickness (b) normalized
error for the week of March 30 – April 5, 2015.





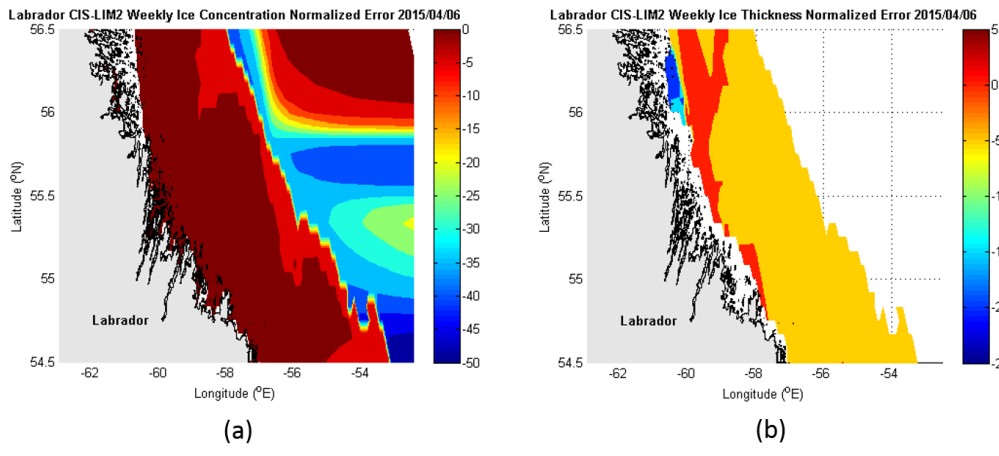


**Figure 8.** Central coastal Labrador weekly average CIS-LIM2 ice concentration (a) and thickness (b) normalized error for the week of April 6-12, 2015.

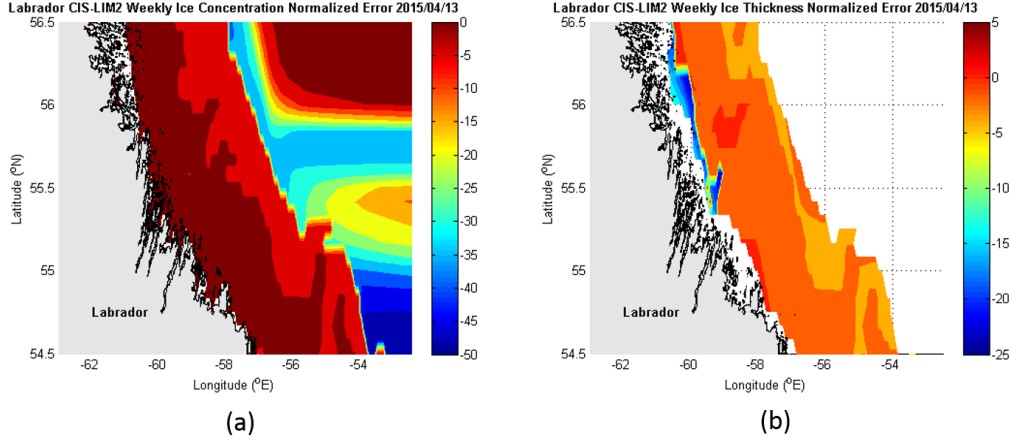


**Figure 9.** Central coastal Labrador weekly average CIS-LIM2 ice concentration (a) and thickness (b) normalized error for the week of April 13-19, 2015.



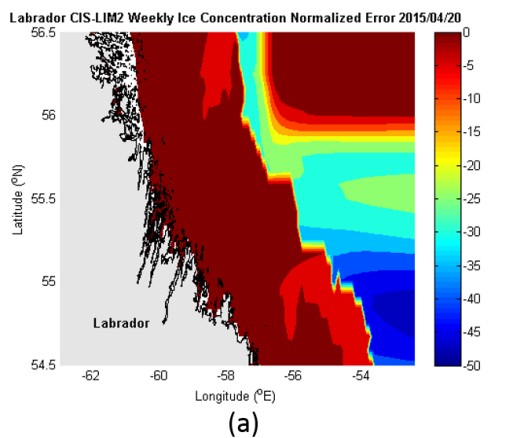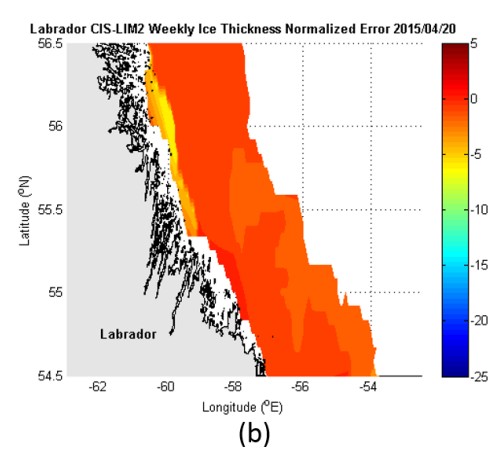


**Figure 10.** Central coastal Labrador weekly average CIS-LIM2 ice concentration (a) and thickness (b) normalized error for the week of April 20-26, 2015.

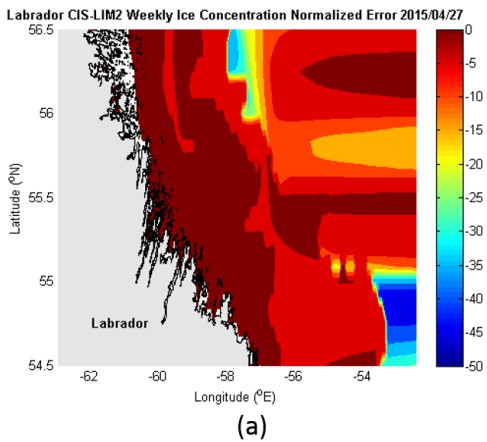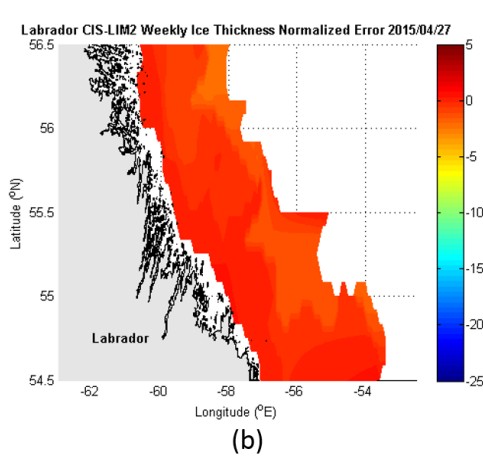


**Figure 11.** Central coastal Labrador weekly average CIS-LIM2 ice concentration (a) and thickness (b) normalized error for the week of April 27 – May 3, 2015.




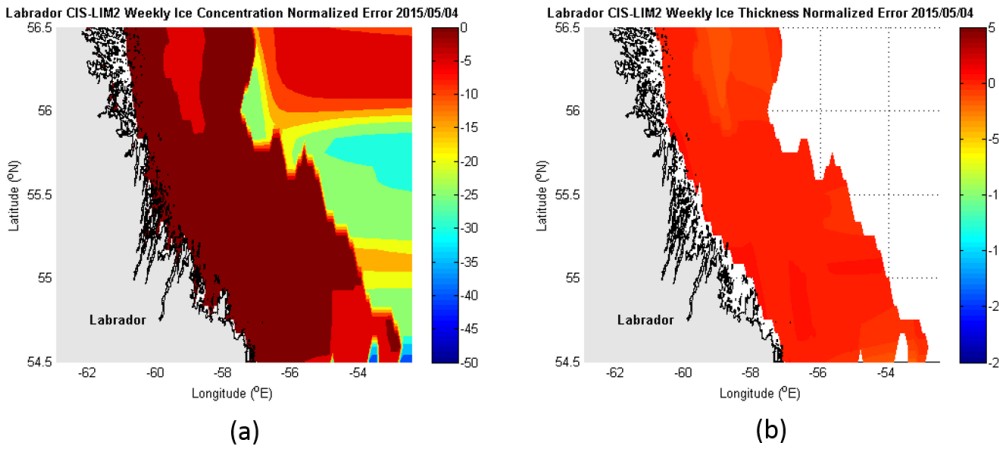


**Figure 12.** Central coastal Labrador weekly average CIS-LIM2 ice concentration (a) and thickness (b) normalized
error for the week of May 4-10, 2015.

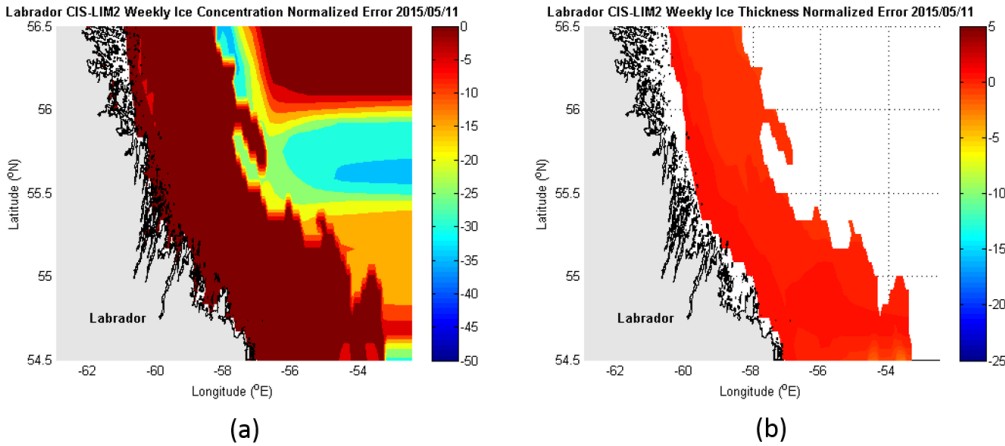


**Figure 13.** Central coastal Labrador weekly average CIS-LIM2 ice concentration (a) and thickness (b) normalized
error for the week of May 11-17, 2015.



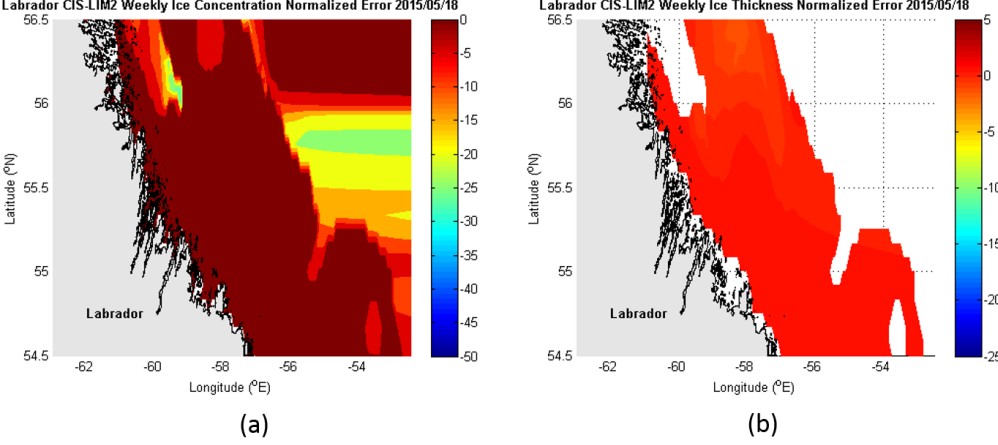

(a)                                                    (b)


**Figure 14.** Central coastal Labrador weekly average CIS-LIM2 ice concentration (a) and thickness (b) normalized error for the week of May 18-24, 2015.

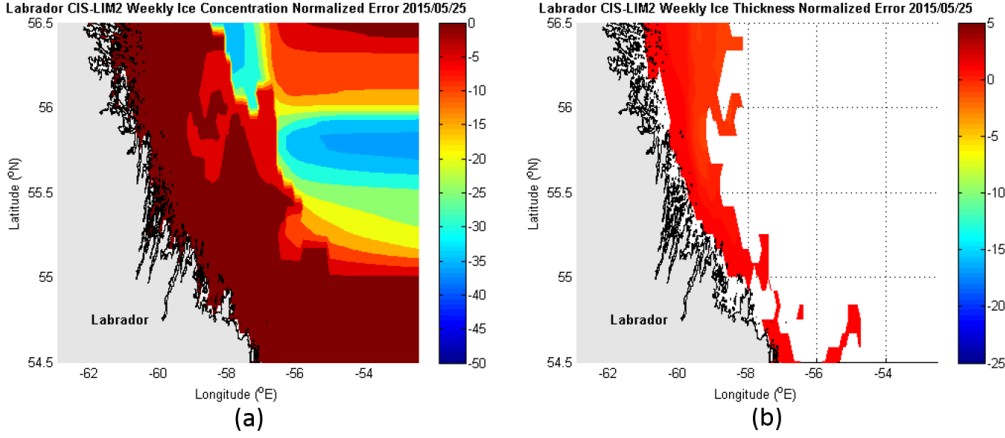

(a)                                                    (b)


**Figure 15.** Central coastal Labrador weekly average CIS-LIM2 ice concentration (a) and thickness (b) normalized error for the week of May 25-31, 2015.

**3 Model Validation**

In this section, results of model runs are presented for the thermodynamic model for April 1 − May 31, 2015, as well as runs of the coupled thermodynamic-dynamic model for May 1-7, 2015. The thermodynamic model run period was selected to cover the observed break-up periods for the land-fast ice offshore Makkovik and Nain. Two of the three ice drift tracking buoys deployed offshore Makkovik began to drift on April 23 (Figure 1), and each of the three buoys deployed offshore Nain began to drift on May 1, 2, and 6, respectively. The shorter thermodynamic-





dynamic model run period was selected to cover most of the observed break-up period for the land-fast ice offshore
Nain.
The model runs are initialized with the best estimate of sea ice conditions over the model domain as obtained from
the LIM2 sea ice model dataset, and model results here are compared with the LIM2 data. Hereafter, all results from
the model presented in this paper are referred to as "model" results, and data from the LIM2 dataset are referred to
explicitly. The model domain extends between 62°W and 57.5°W longitude and 55°N and 57°N latitude. For the
thermodynamic model-only runs, the parameters of interest which are presented here are the sea ice thickness and
concentration. For the thermodynamic-dynamic model, the sea ice thickness, concentration, and velocity are
presented. Of particular interest in the present work is the model performance in the regions offshore Makkovik and
Nain; hence, additional analyses are given for the temporal evolution of sea ice thickness and concentration in these
areas.
The thermodynamic model runs use a 0.45° spatial resolution (approximately 50 km) for initial ice particle spacing
and a one-hourly temporal resolution, while the thermodynamic-dynamic model run uses a 0.5° spatial resolution
(approximately 55.5 km) for initial ice particle spacing and a one-minute temporal resolution. The higher temporal
resolution is needed for the stability of the numerical integration scheme in the dynamic model. Hence, a lower
spatial resolution and shorter model run period are used for the dynamic model to lessen the required computation
time.

### 3.1 Thermodynamic model results

The largest uncertainty in the thermodynamic model is the initialization of snow cover on the sea ice. While
available reanalysis datasets provide snow depth, they only provide it in meters of water equivalent; however, actual
snow depth on the sea ice plays a critical role in the evolution of the ice cover due to the strong insulating effect of
snow. Since there do not exist any widespread and frequent snow depth measurements along the Labrador coast, an
approximate early spring mean snow depth value of 40 cm is used here to initialize the model on April 1. This depth
was taken from measurements of snow depth performed on Lake Melville in March 2009 (*e.g.*, Prinsenberg et al.,
2011). It cannot be overstated how uncertain this initial snow depth value is, as it is applied to the entire model
domain, and snow depths may be significantly greater further north around Nain. Future use of this model for
operational ice forecasting purposes must involve adequate snow depth data acquisition.
In the model results presented here, one model run assumes 40 cm of snow depth over the entire model domain at
the start of April 1, 2015, while another model run assumes no snow on the sea ice at the start of the model run on
April 1. Both model runs are subject to the same snow accumulation on the ice over the two-month period as forced
by the ECMWF snowfall data. Hence, the effect of a significant snow cover on the ice at the start of the melt season
can be examined.
Figure 16 shows the initial ice conditions on April 1, 2015 00:00 UTC as given by the LIM2 dataset. The green
circle represents the region offshore Makkovik analyzed here, and the purple circle represents the region offshore
Nain. There are four model grid points within each of these circled regions over which ice thickness and





concentrations are averaged for the offshore Makkovik and Nain analyses. These two regions were selected to
encompass the land-fast ice in the vicinities of Makkovik and Nain so that the timing of the land-fast ice break-up
could be modeled and compared with the actual timing of the break-up as recorded by the ice tracking buoys.

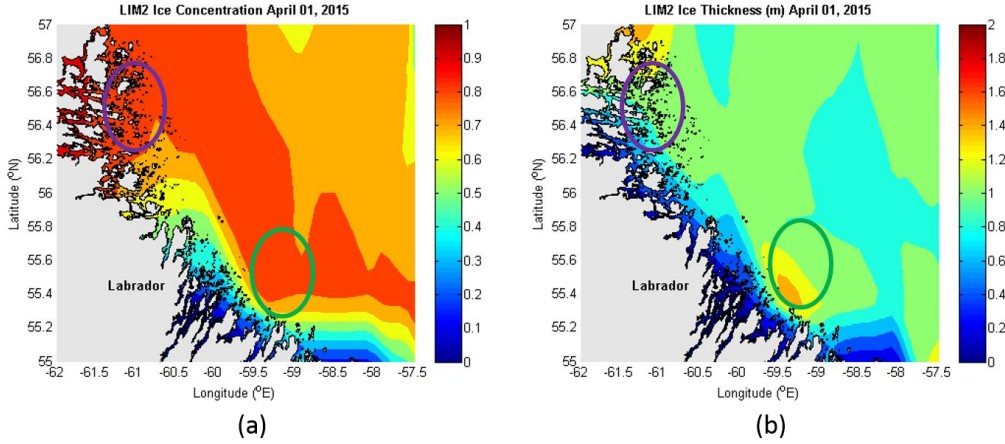


**Figure 16.** Sea ice concentration (a) and thickness (b) over the model domain at the start of the thermodynamic

model runs on April 1, 2015 00:00 UTC, as obtained from the LIM2 dataset. The region circled in green represents
the area over which ice conditions offshore Makkovik are analyzed in this section, and the purple circled region
represents the area over which ice conditions offshore Nain are analyzed.
Figure 17 shows the evolution of average ice conditions during April 1 – May 31, 2015 in the regions offshore
Makkovik (green circles in Figure 16) and Nain (purple circles in Figure 16) according to the LIM2 data, the
thermodynamic model with 40 cm of initial snow cover, and the thermodynamic model with no initial snow cover.
According to the model run with initial snow cover, the average ice thickness and concentration offshore Makkovik
decrease to zero on May 14, while for the run with no initial snow cover, the ice disappears completely around
Makkovik on May 1. The progression of the LIM2 ice thickness and concentration near Makkovik shows a
somewhat less precipitous decline in ice conditions compared to the model results shown here. While the LIM2 ice
thickness and concentration do not reach zero during the model time period, they decline to less than 0.2 m and 0.1,
respectively, by May 18 before rebounding slightly as more ice drifts into the region from the north. This minimum
in overall average ice conditions offshore Makkovik occurs four days after the model minimum with 0.4 m initial
snow cover, and 18 days after the disappearance of ice in the model run with no initial snow cover. In the region
offshore Nain, the ice also disappears on May 14 in the model with initial snow cover, and on April 25 in the model
without initial snow cover (Figure 17). The LIM2 ice thickness and concentration decrease to zero by May 15.



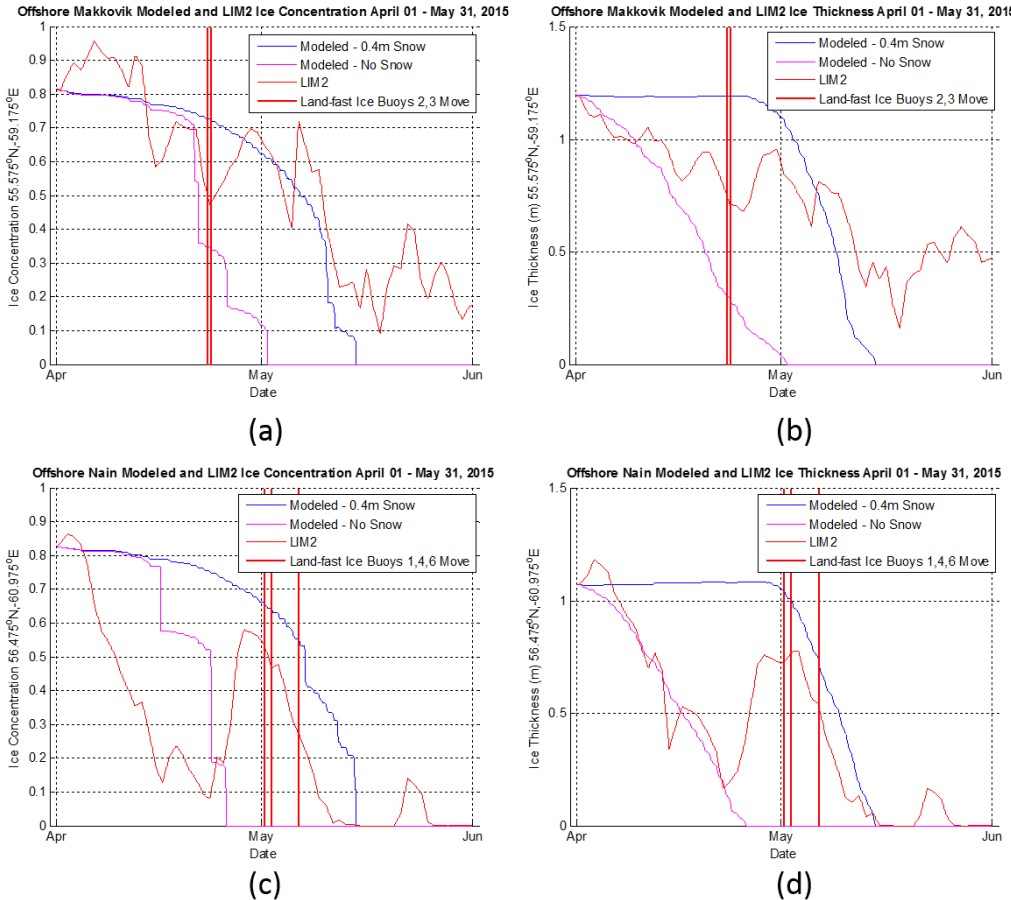

(a)  (b)

(c)  (d)


**Figure 17.** Evolution of average ice concentration (a,c) and thickness (b,d) in the focus regions offshore Makkovik (a,b) and Nain (c,d) during April 1 – May 31, 2015. The thin red lines represent LIM2 ice conditions, the blue lines represent ice conditions for the thermodynamic model presented here initialized with 40 cm of snow, and the magenta lines represent ice conditions for the thermodynamic model initialized with no snow. The thick red lines mark the timing of first recorded movement by the land-fast ice buoys. The latitude-longitude coordinates shown on the y-axes represent the central locations for the regions over which average ice conditions are shown.

The thick red lines in Figure 17 mark the times that the land-fast ice buoys first recorded movement, indicating that the break-up of the land-fast ice in those regions had begun. Buoys 2 and 3 began to drift on April 23 around 05:00 and 15:00 UTC, respectively, offshore Makkovik, and buoys 1, 4, and 6 began to drift around May 1 10:00, May 2 12:00, and May 6 12:00 UTC, respectively, offshore Nain. The model run with 40 cm initial snow cover most accurately predicts the timing of the onset of the land-fast ice break-up with precipitous declines in regional mean ice thickness and concentration commencing immediately after these times. The averaged results for both the Makkovik and Nain regions suggest that there was most likely significant snow cover on the ice on April 1, and that





if the model is properly initialized with snow cover conditions, it can reasonably predict the progression and timing
of the decline in regional ice concentration and thickness more than a month in advance.
At the specific locations at which the ice drift buoys were deployed, Figure 18 shows the modeled ice concentrations
and thicknesses near Makkovik and Nain for April 1 – May 31 from the thermodynamic model with 0.4 m of initial
snow cover. This model run is not initialized with the LIM2 ice concentration and thickness data, but rather assumes
the land-fast ice at these five locations had a concentration and thickness of 0.995 and 0.95 m on April 1,
respectively. The assumed ice thickness represents the mean of the normal thickness range for medium first-year ice
(0.7-1.2 m), which was the ice type shown for these locations on April 1 on the CIS daily ice charts. In Figure 18,
the times at which the modeled ice concentrations first drop below 0.99 and the modeled thicknesses first began to
display steep decreases below their original thickness of 0.95 m are used as proxies for the beginning of the local of
land-fast ice break-up. These times fall within 4.7 hours to 5.9 days from the times at which the drift buoys began to
move at these locations (Table 4).



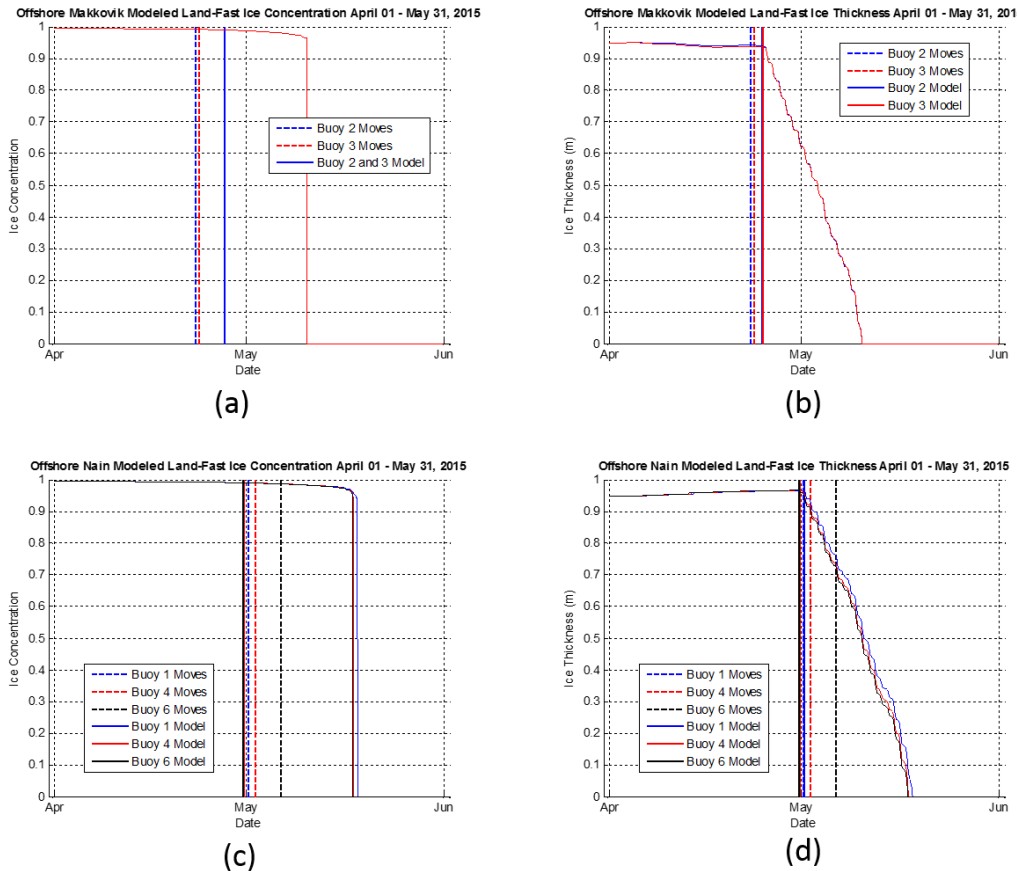


**Figure 18.** Modeled ice concentration and thickness (thin solid lines) vs. time at the buoy deployment locations near Makkovik (a,b) and Nain (c,d). The vertical dashed lines represent the times at which the buoys first recorded motion, the vertical solid lines represent the times at which the modeled ice concentrations begin to fall below 0.99 (a,c), and the times at which the modeled ice thicknesses first fall below 0.95 m (all times in UTC).

**Table 4.** Summary of buoy deployment locations, modeled and observed land-fast ice break-up times according to concentration and thickness criteria described above, and observed minus modeled break-up time error ranges.

| Buoy | Deployment Coordinates | Modeled Break-Up Time (Thickness) | Modeled Break-Up Time (Concentration) | Observed Break-Up Time | Observed - Modeled Error Range (hours) |
|---|---|---|---|---|---|
| 1 | 56.45°N, 60.78°W | May 01 14:00 | April 30 17:00 | May 01 09:18 | -4.7 to 16.3 |





| 2 | 55.29°N, 59.24°W | April 24 21:00 | April 27 14:00 | April 23 04:32 | -105.5 to -40.5 |
|---|---|---|---|---|---|
| 3 | 55.28°N, 59.19°W | April 25 02:00 | April 27 14:00 | April 23 14:14 | -95.8 to -35.8 |
| 4 | 56.40°N, 60.80°W | April 30 20:00 | April 30 16:00 | May 02 11:26 | 39.4 to 43.4 |
| 6 | 56.39°N, 60.84°W | April 30 18:00 | April 30 15:00 | May 06 11:38 | 137.6 to 140.6 |

Figure 19 shows contour plots of the modeled and LIM2 ice thickness and concentration across the model domain
24 days into the model run, and one day after the first land-fast ice buoys offshore Makkovik began to drift. The
model results shown in Figure 19 are from the run initialized with 0.4 m of snow cover. The model shows little
change in ice thicknesses by April 24 compared to April 1, with some decrease in ice concentrations. The LIM2 data
show a comparatively more extensive retreat in ice thicknesses and concentrations by April 24; however, both the
LIM2 data and the model results show a large region of ice concentrations around 0.7 toward the coast, with
concentrations around 0.6 immediately to the east.





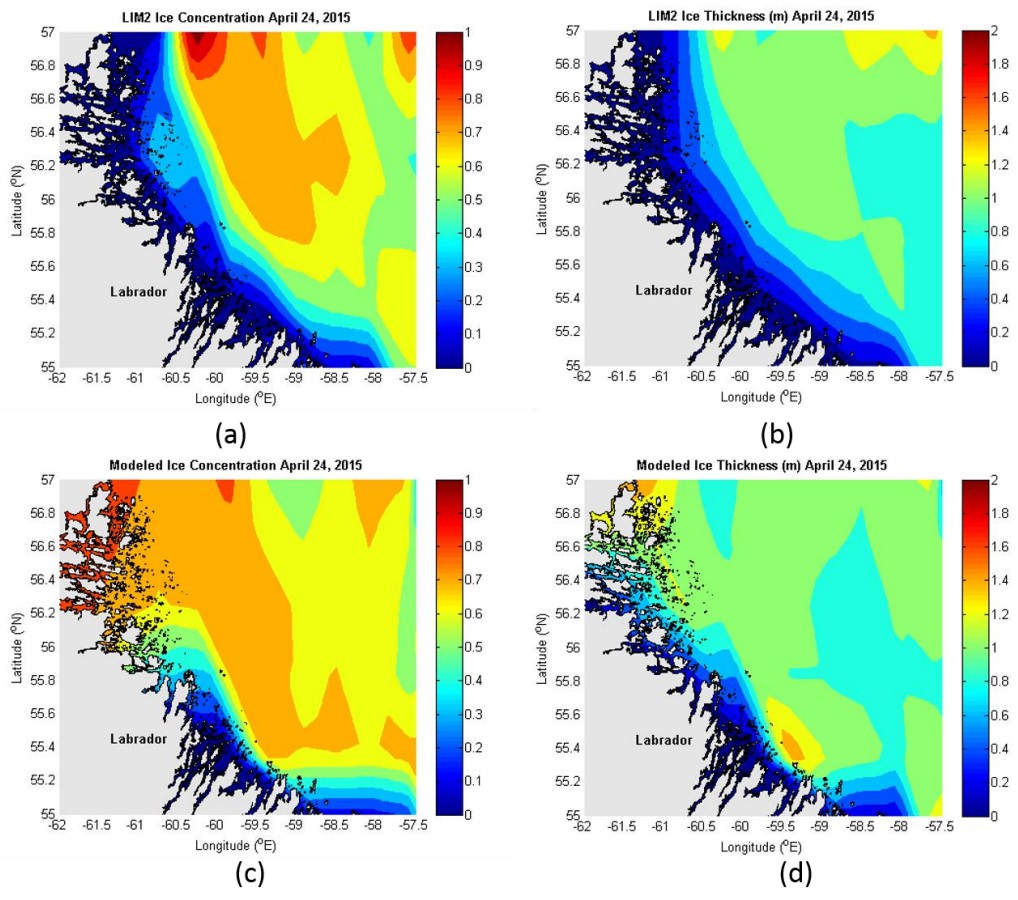

Figure 19. Sea ice concentration (a,c) and thickness (b,d) on April 24, 2015 03:00 UTC (00:00 AST) from LIM2 data (a,b) and the thermodynamic model presented here (c,d). The results of the thermodynamic model run shown here are initialized with 40 cm of snow.

The results in Figure 20 show that when the thermodynamic model is initialized with no snow cover on April 1, nearly all the ice disappears by April 24. In this model run, the ice disappears completely from the model domain by May 2, which indicates that it is not a realistic scenario to initialize the model without a snow cover.



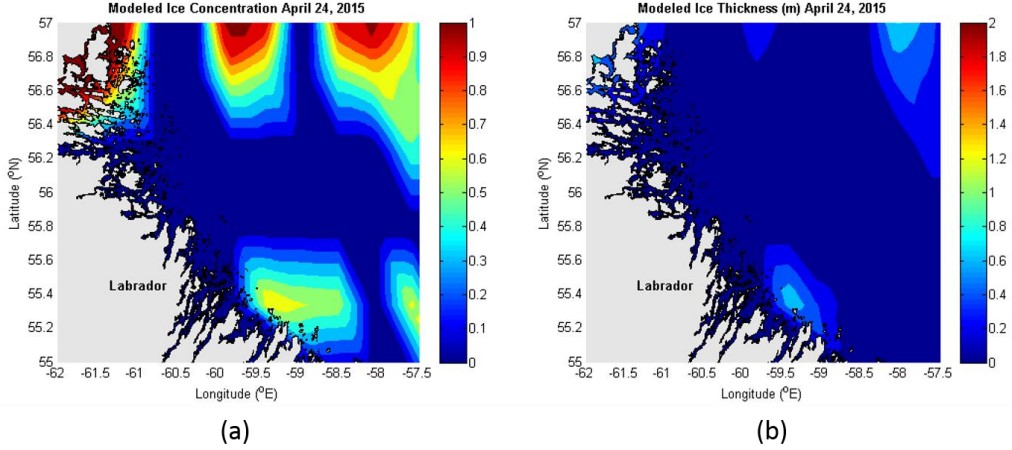

773

**Figure 20.** Sea ice concentration (a) and thickness (b) on April 24, 2015 03:00 UTC (00:00 AST) from the thermodynamic model with no initial snow cover.

Figure 21-Figure 23 show contour plots of the modeled and LIM2 ice thickness and concentration across the model domain for May 2, 3, and 7, respectively. These dates were selected to coincide with the timing of the first detected motion of the land-fast ice buoys deployed offshore Nain. These buoys began to drift on May 1, 2, and 6. The model results shown here are all from the run initialized with 0.4 m of snow cover. While the modeled results and LIM2 data show somewhat different patterns of ice retreat during the May 2-7 period, both show a general pattern of the retreat of thicker, more concentrated ice toward the northern end of the model domain.



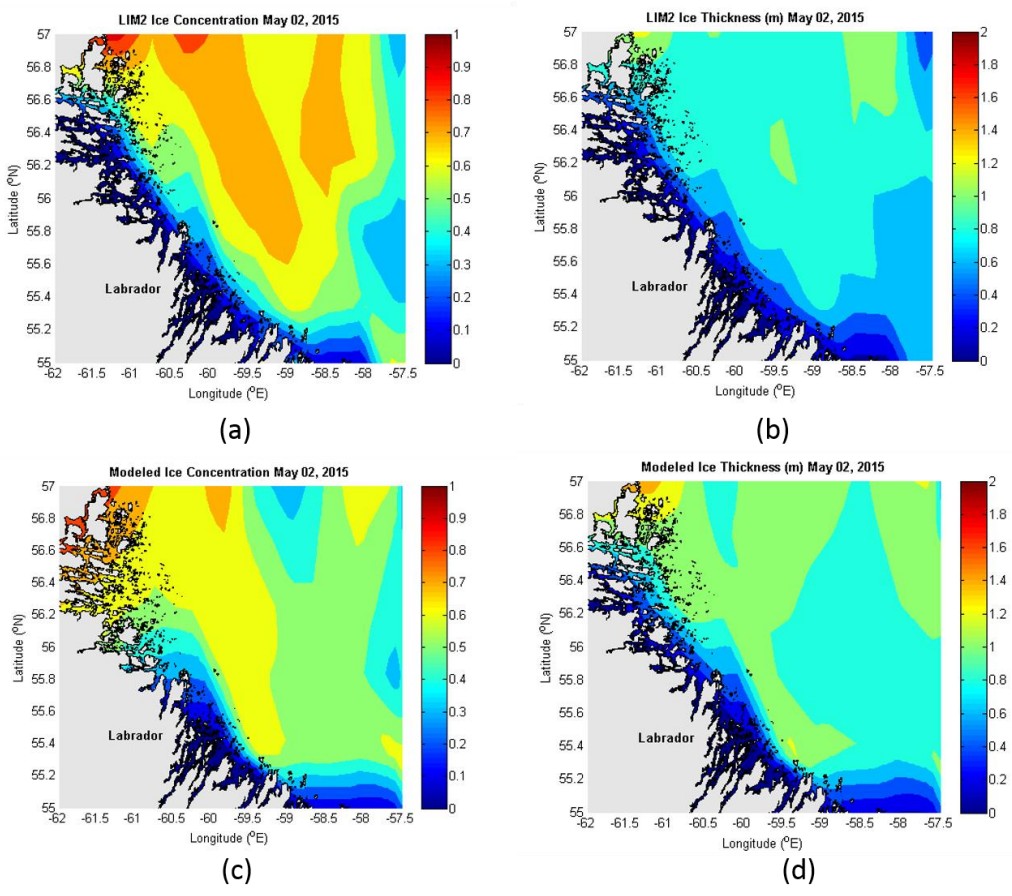

**Figure 21.** Sea ice concentration (a,c) and thickness (b,d) on May 02, 2015 03:00 UTC (00:00 AST) from LIM2 data (a,b) and the thermodynamic model presented here (c,d). The results of the thermodynamic model run shown here are initialized with 40 cm of snow.





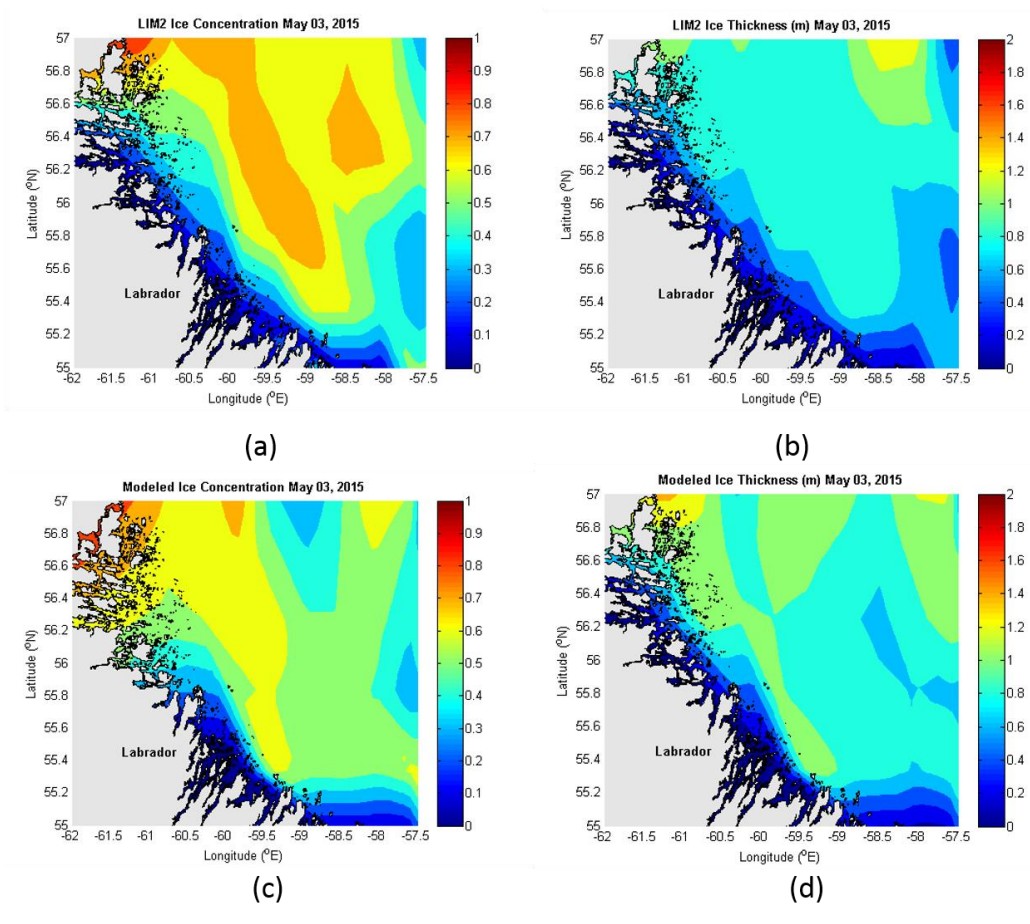

786

**Figure 22.** Sea ice concentration (a,c) and thickness (b,d) on May 03, 2015 03:00 UTC (00:00 AST) from LIM2
data (a,b) and the thermodynamic model presented here (c,d). The results of the thermodynamic model run shown
here are initialized with 40 cm of snow.





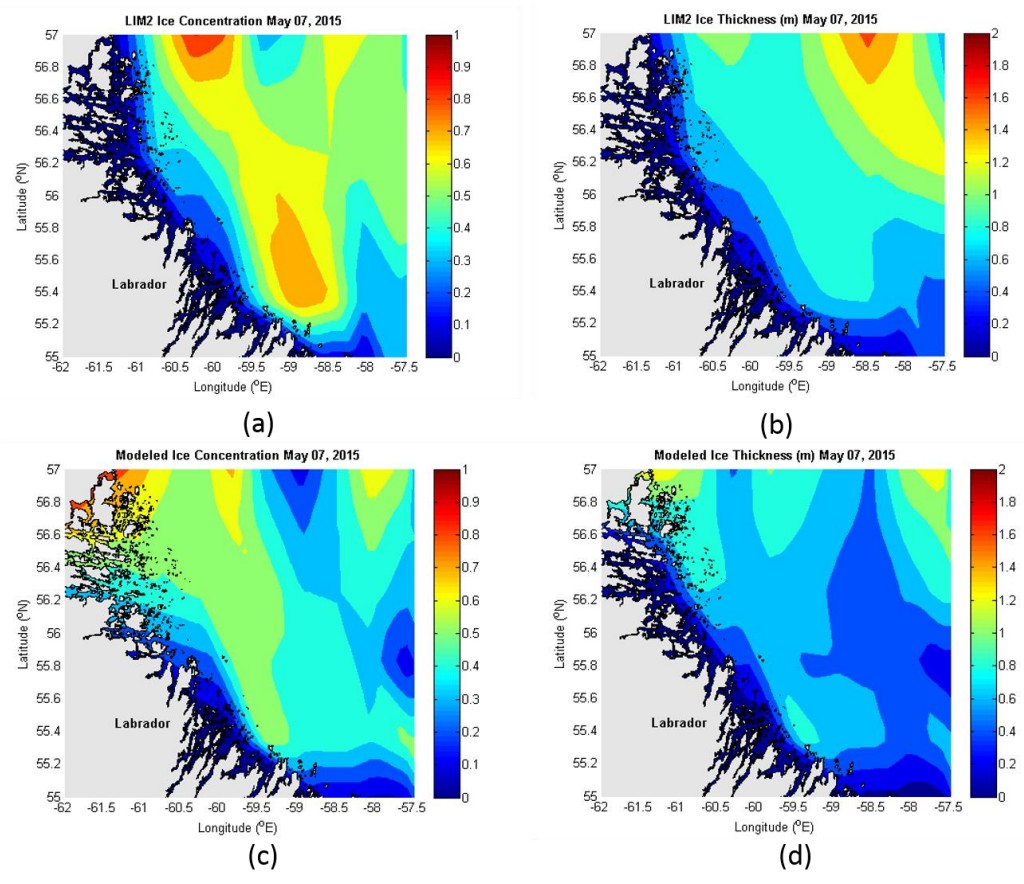

(a)                                    (b)

(c)                                    (d)

790

**Figure 23.** Sea ice concentration (a,c) and thickness (b,d) on May 07, 2015 03:00 UTC (00:00 AST) from LIM2 data (a,b) and the thermodynamic model presented here (c,d). The results of the thermodynamic model run shown here are initialized with 40 cm of snow.

Figure 24 shows the evolution of average ice conditions during April 1 – May 31, 2015 over the model domain according to the LIM2 data, the thermodynamic model with 40 cm of initial snow cover, and the thermodynamic model with no initial snow cover. The spatially averaged ice thickness and concentration for the model run with 0.4 m of initial snow cover tracks closely to the LIM2 mean thickness and concentration, and begins to decrease significantly on May 11 when both the mean thickness and concentration decrease to less than 0.5 (m or concentration). The ice disappears completely from the model domain on May 15. The spatially averaged ice thickness and concentration for the model run with no initial snow cover tracks closely to the LIM2 mean thickness and concentration, and begins to decrease significantly on April 21 when both the mean thickness and concentration decrease to less than 0.5 (m or concentration). In the model run with no initial snow cover, the ice disappears completely on May 1. The LIM2 mean domain ice thickness and concentration both remain fairly constant around




0.6 (m or concentration) during April 1 – May 11, and subsequently begin to display greater variability between 0.3-
0.7 (m or concentration) while showing an overall decrease, until May 31.

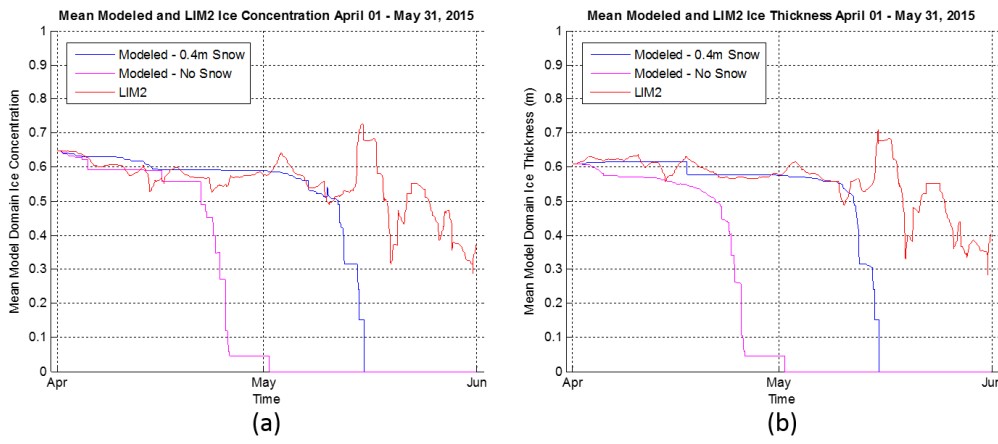


**Figure 24.** Evolution of average ice concentration (a) and thickness (b) over the entire model domain during April 1
– May 31, 2015. The red lines represent LIM2 ice conditions, the blue lines represent ice conditions for the
thermodynamic model presented here initialized with 40 cm of snow, and the magenta lines represent ice conditions
for the thermodynamic model initialized with no snow.
Table 5 summarizes the model domain mean ice thicknesses and concentrations for the selected dates shown in
Figure 19-Figure 23 for the LIM2 dataset, the model run with 0.4 m of initial snow cover, and the model run with no
initial snow cover.
**Table 5.** Summary of modeled vs. LIM2 model domain mean ice concentration and thickness for selected dates for
April 1 – May 31, 2015 model run.

| Modeled vs. LIM2 Mean Domain Ice Concentration and Thickness (Thermodynamic Model) | | | | | |
|---|---|---|---|---|---|
| **Date** | **April 01 Snow Cover (m)** | **LIM2 Concentration** | **Modeled Concentration** | **LIM2 Thickness (m)** | **Modeled Thickness (m)** |
| **April 24** | 0.40 | 0.42 | 0.57 | 0.43 | 0.48 |
| | 0.00 | | 0.44 | | 0.43 |
| **May 02** | 0.40 | 0.77 | 0.54 | 0.65 | 0.47 |
| | 0.00 | | 0.00 | | 0.00 |
| **May 03** | 0.40 | 0.78 | 0.53 | 0.69 | 0.47 |
| | 0.00 | | 0.00 | | 0.00 |



| | 0.40 | | 0.46 | | 0.45 |
|---|---|---|---|---|---|
| **May 07** | | 0.36 | | 0.39 | |
| | 0.00 | | 0.00 | | 0.00 |


### 3.2 Thermodynamic-dynamic model results

The coupled thermodynamic-dynamic model run for May 1-7, 2015 is initialized with the LIM2 ice thickness,
concentration, and velocity grid for May 1 00:00 UTC (Figure 25). The green circle represents the single coordinate
offshore Makkovik analyzed here (55.25°N, 59.25°W), and the purple circle represents the coordinate offshore Nain
(56.5°N, 61°W). Given that the ice particles are free to drift from their initial positions in the dynamic model, the
modeled ice conditions at these coordinates are linearly interpolated.

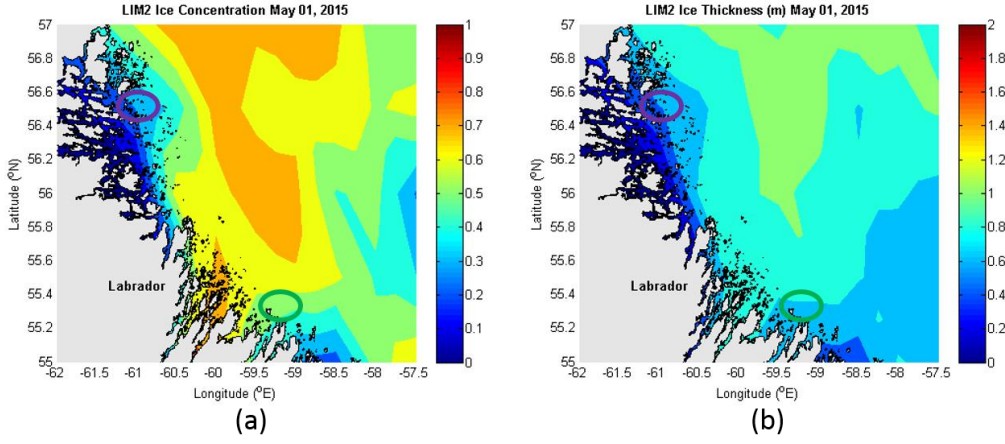


**Figure 25.** Sea ice concentration (a) and thickness (b) over the model domain at the start of the thermodynamic-
dynamic model run on May 1, 2015 00:00 UTC, as obtained from the LIM2 dataset. The region circled in green
represents the area over which ice conditions offshore Makkovik are analyzed in this section, and the purple circled
region represents the area over which ice conditions offshore Nain are analyzed.
Figure 26 shows the evolution of modeled and the LIM2 ice thickness (a,c) and concentration (b,d) during May 1-7
offshore Makkovik (a,b) and Nain (c,d).





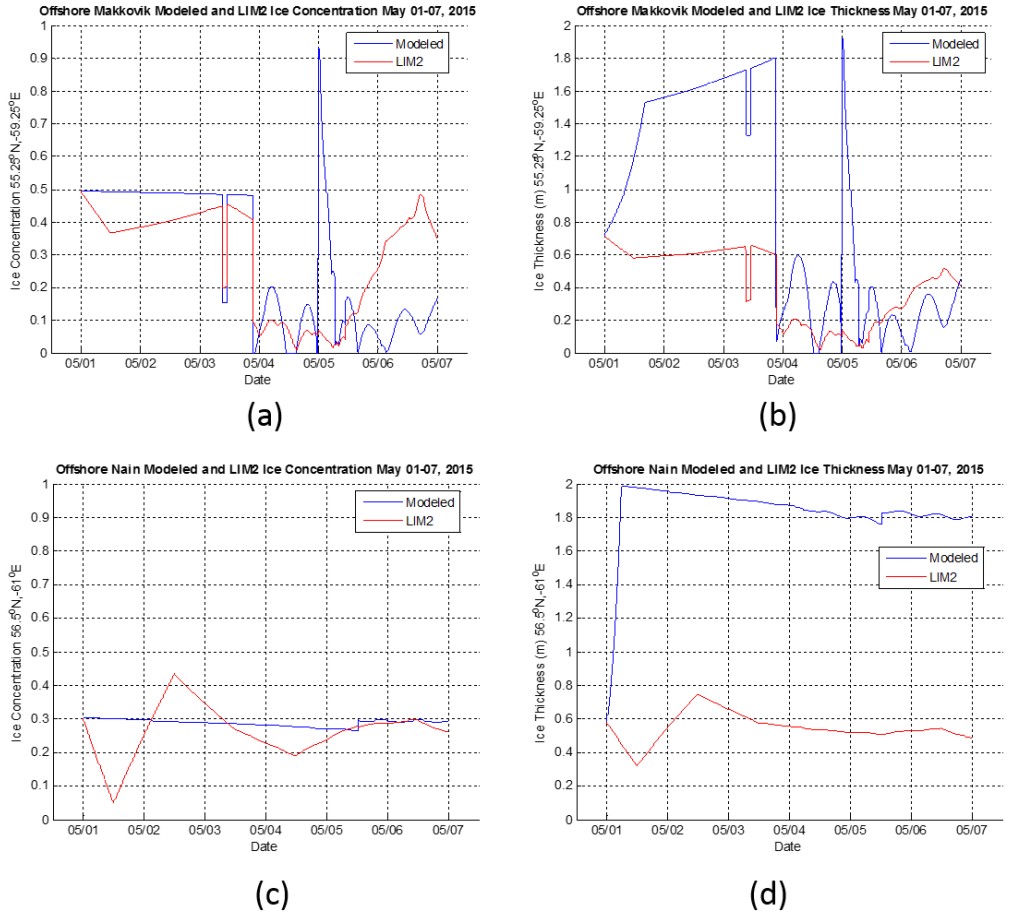

**Figure 26.** Evolution of average ice concentration (a,c) and thickness (b,d) in the focus regions offshore Makkovik (a,b) and Nain (c,d) during May 1-7, 2015. The red lines represent LIM2 ice conditions, and the blue lines represent ice conditions for the thermodynamic-dynamic model. The latitude-longitude coordinates shown on the y-axes represent the central locations for the regions over which average ice conditions are shown.

Figure 27-Figure 30 show the modeled ice drift velocities and total speeds along the trajectories of buoys 1-4 during May 1-7. The plots for buoy 4 begin on May 2 when the buoy began to record drift (Figure 30). Results for buoys 5-6 are not shown because buoy 5 remained stationary on the land-fast ice offshore Makkovik for its entire transmission period, and buoy 6 did not drift for more than 24 hours during the model run period. The results show that the model can reproduce mean regional ice drift velocities; however the model tends to underestimate high-frequency spikes in drift speeds compared to those recorded by the drift buoys 1 and 4 (Figure 27 and Figure 30). Along the track of buoy 1, the model does not capture the tidally-forced oscillations in the ice drift velocity as recorded by the buoy; this is most likely due to the inadequate spatial resolution of the NEMO 3.1 ocean current




input. For the along-track drift speeds of buoys 2 and 3, the model overestimates the increase in ice drift speeds
during May 3-7 (Figure 28-Figure 29).

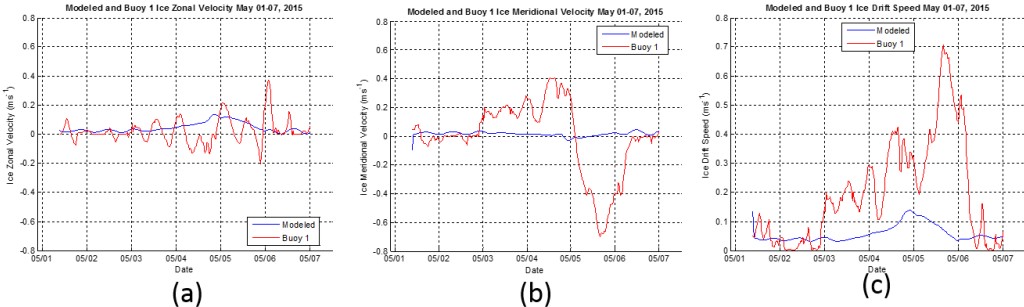


**Figure 27.** Modeled (blue lines) and buoy 1 measured (red lines) along-track ice drift zonal (a), meridional (b)
velocities, and speeds (c) during May 1-7, 2015 offshore Labrador.

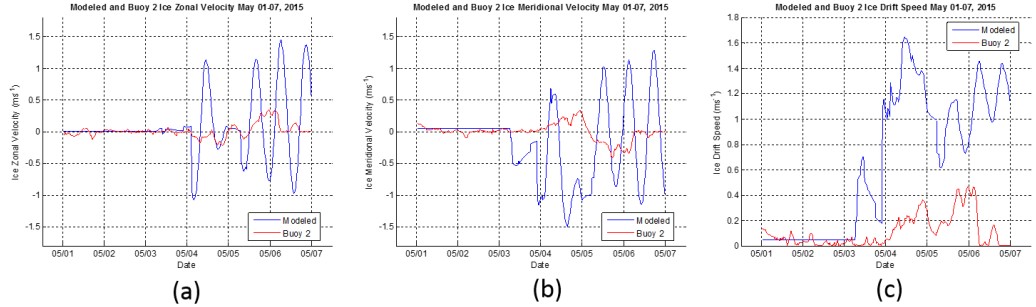


**Figure 28.** Modeled (blue lines) and buoy 2 measured (red lines) along-track ice drift zonal (a), meridional (b)
velocities, and speeds (c) during May 1-7, 2015 offshore Labrador.

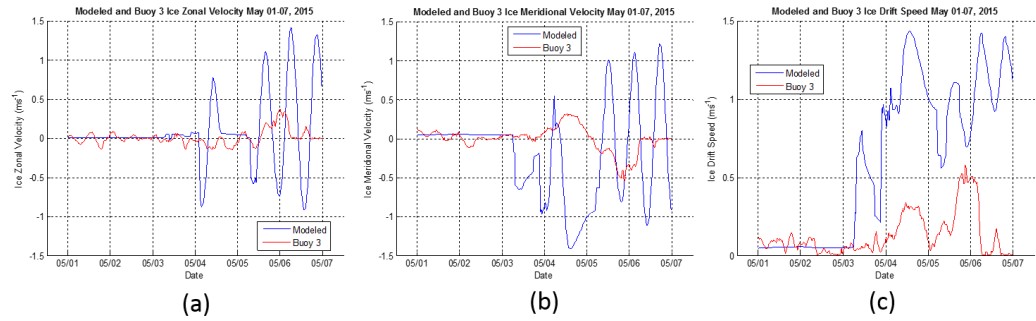


**Figure 29.** Modeled (blue lines) and buoy 3 measured (red lines) along-track ice drift zonal (a), meridional (b)
velocities, and speeds (c) during May 1-7, 2015 offshore Labrador.





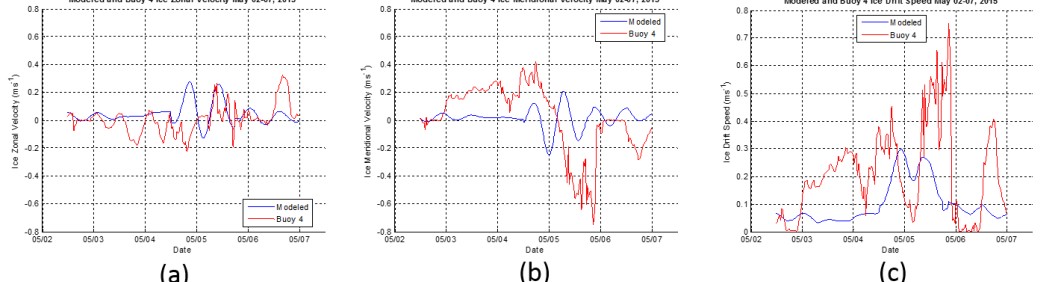


**Figure 30.** Modeled (blue lines) and buoy 4 measured (red lines) along-track ice drift zonal (a), meridional (b) velocities, and speeds (c) during May 1-7, 2015 offshore Labrador.

Figure 31 shows the observed minus modeled ice drift speed errors during May 1-7 for buoys 1-4. The buoy-recorded and modeled drift speeds were very close during May 1-3. For buoys 1 and 4, the observed-modeled drift speed error increases to 0.2-0.5 m s$^{-1}$ during May 3-7. For buoys 2 and 3, the observed-modeled drift speed error increases to 0.5-1.5 m s$^{-1}$ during this same period. These results indicate that the model is better at simulating observed ice drift speeds further offshore compared to nearshore drift speeds, as buoys 1 and 4 drifted further offshore than buoys 2 and 3 (see Figure 1). Beginning on May 3, the modeled ice velocities along the tracks of buoys 2 and 3 increase significantly over the observed buoy-tracked ice velocities. This is due to increases in the internal ice stresses, sea surface tilt forcing, and coastal boundary forcing along the nearshore tracks of buoys 2 and 3 in the model.

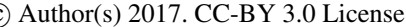



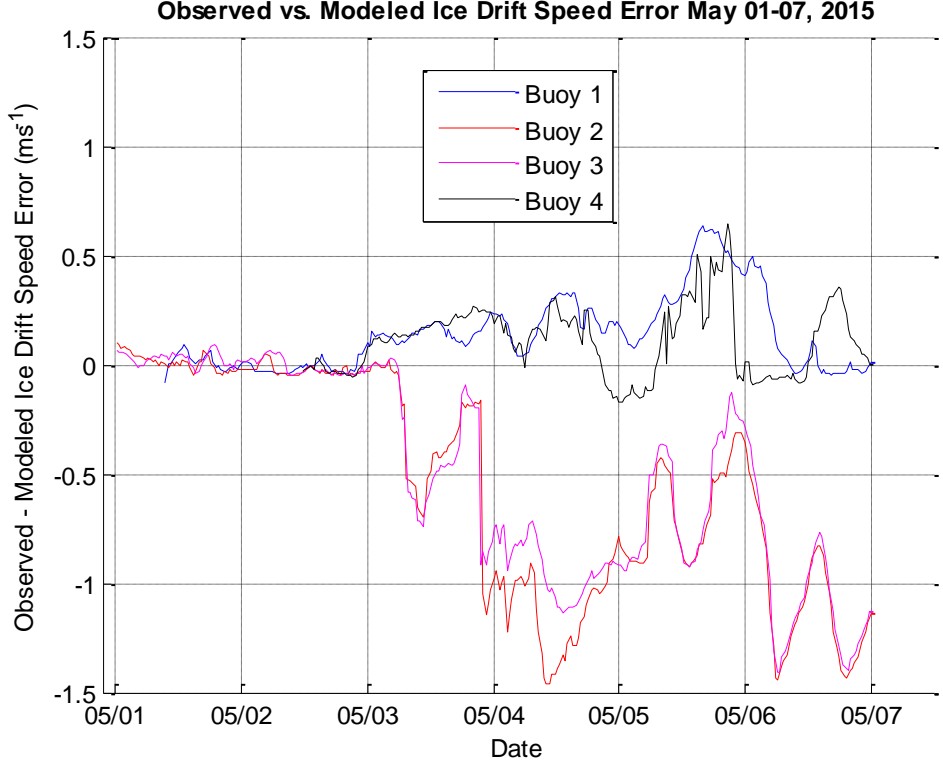


**Figure 31.** Observed minus modeled ice drift speeds along the trajectories of buoys 1-4 during May 1-7, 2015

offshore Labrador.
Figure 32-Figure 34 show the LIM2 and modeled ice thickness and concentrations for the model domain for May 3,
5, and 7, respectively. The model tends to overestimate ice thicknesses and concentrations across the domain;
however the model does capture the fact that a tongue of higher ice concentrations develops toward the middle of
the model domain.





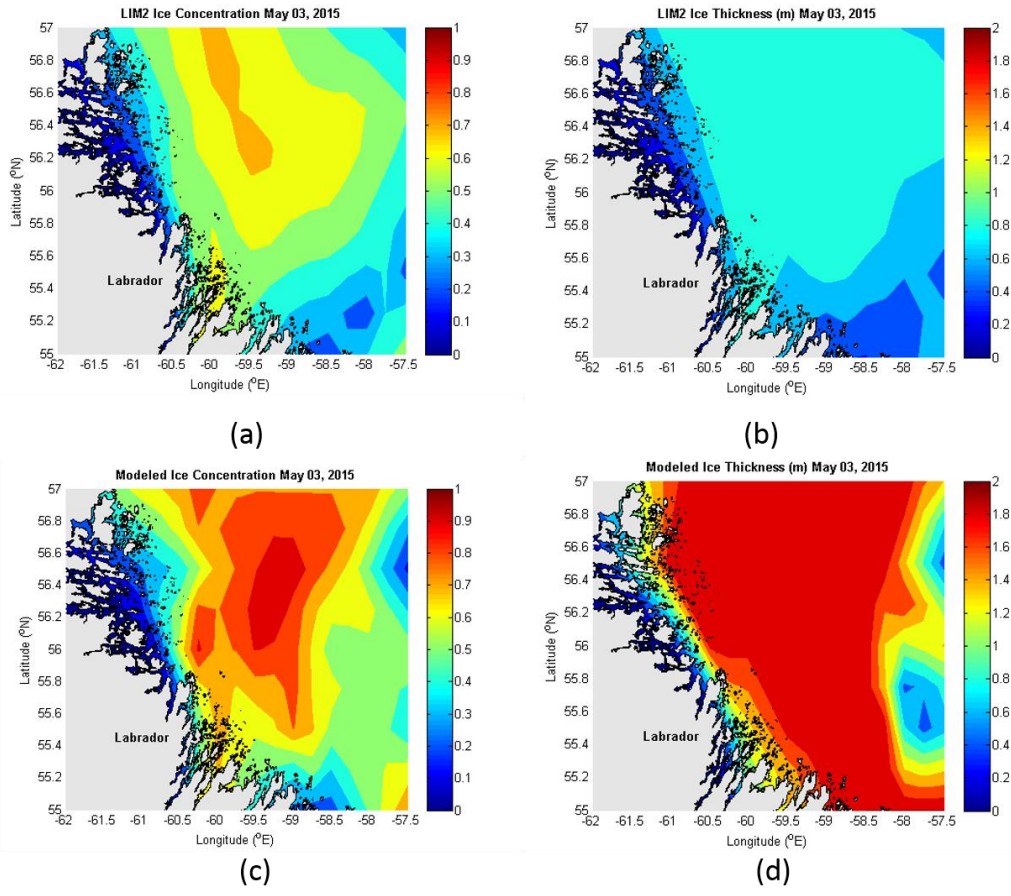

**Figure 32.** Sea ice concentration (a,c) and thickness (b,d) on May 03, 2015 03:00 UTC (00:00 AST) from LIM2 data (a,b) and the thermodynamic-dynamic model presented here (c,d).



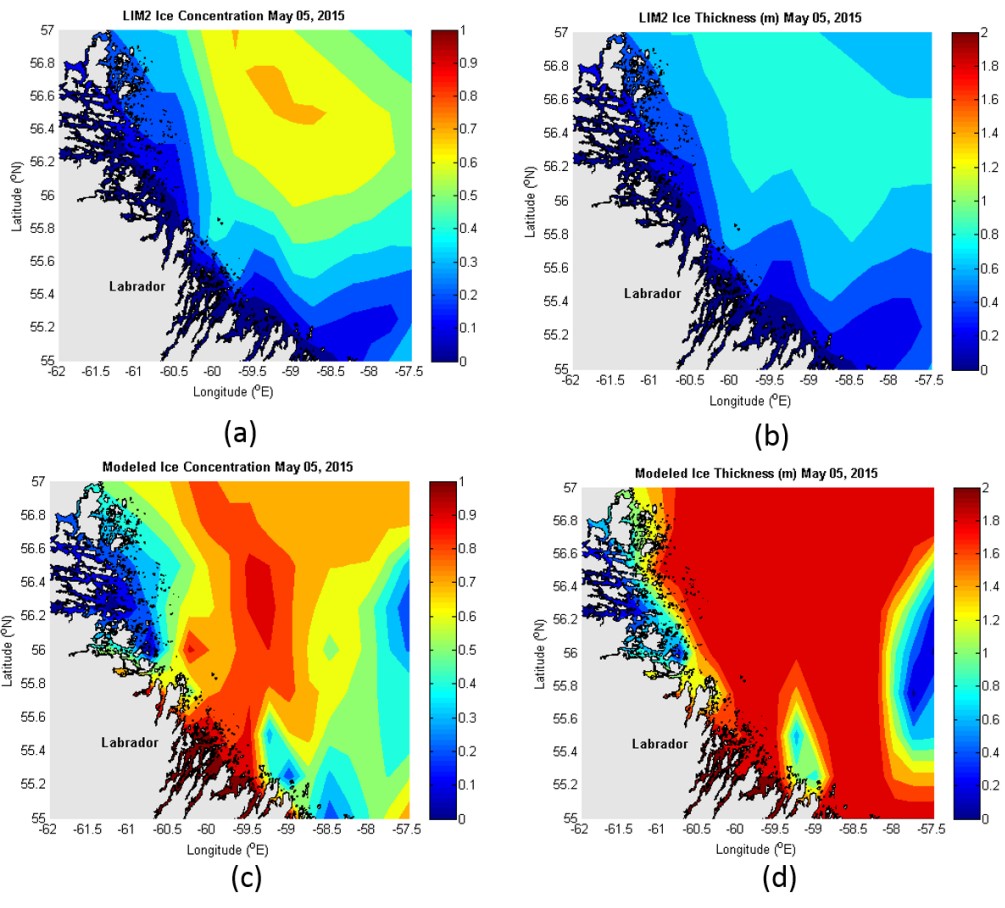


**Figure 33.** Sea ice concentration (a,c) and thickness (b,d) on May 05, 2015 03:00 UTC (00:00 AST) from LIM2
data (a,b) and the thermodynamic-dynamic model presented here (c,d).



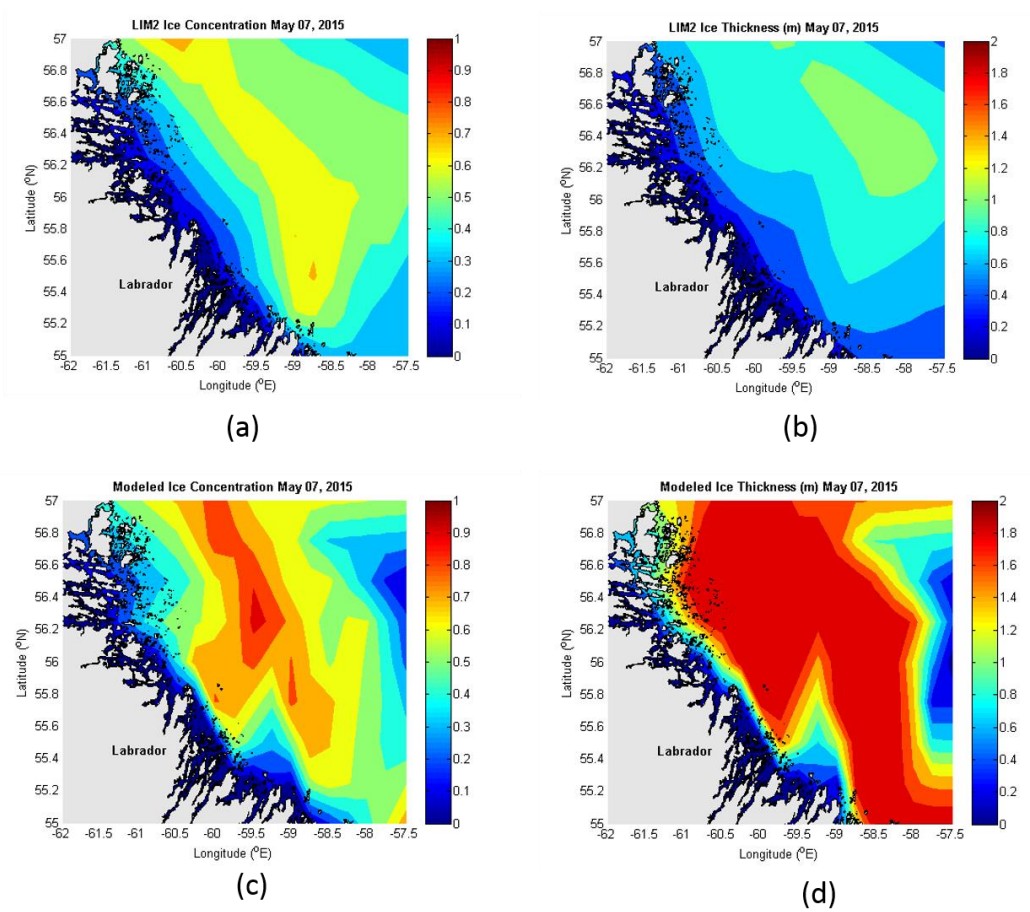


**Figure 34.** Sea ice concentration (a,c) and thickness (b,d) on May 07, 2015 00:00 UTC (May 06 21:00 AST) from
LIM2 data (a,b) and the thermodynamic-dynamic model presented here (c,d).

Figure 35 shows the evolution of the mean model domain ice thickness (a) and concentration (b) during May 1-7 for
the model and the LIM2 dataset.



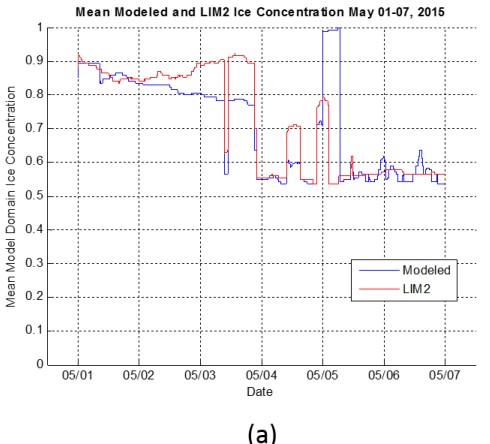
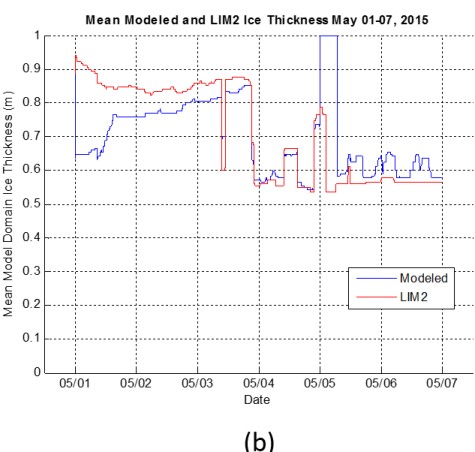

(a)                (b)


**Figure 35.** Evolution of average ice concentration (a) and thickness (b) over the entire model domain during May 1-
7, 2015. The red lines represent LIM2 ice conditions, and the blue lines represent ice conditions for the
thermodynamic-dynamic model presented here.
Table 6 summarizes the mean model domain ice thicknesses and concentrations for the model and the LIM2 dataset
for the selected dates analyzed in this section.
**Table 6.** Summary of modeled vs. LIM2 model domain mean ice concentration and thickness for selected dates for
May 1-7, 2015 model run.

| Modeled vs. LIM2 Mean Domain Ice Concentration and Thickness (Thermodynamic-Dynamic Model) | | | | |
|---|---|---|---|---|
| **Date** | **LIM2 Concentration** | **Modeled Concentration** | **LIM2 Thickness (m)** | **Modeled Thickness (m)** |
| **May 03** | 0.55 | 0.78 | 0.54 | 0.81 |
| **May 05** | 0.42 | 0.75 | 0.42 | 0.80 |
| **May 07** | 0.39 | 0.55 | 0.38 | 0.79 |

**4 Conclusions**
A model is presented in this paper which can be used to generate short-term (several days) to seasonal (months)
forecasts of regional ice conditions for coastal Labrador. It was shown that when the thermodynamic model was
used by itself for the April-May 2015 period, forecasts of the timing of the near-disappearance of ice near Makkovik
and Nain, Labrador were accurate to within one to four days when the model was initialized on April 1 (Figure 17).
For the specific locations on the land-fast ice at which the tracking buoys were originally deployed, the
thermodynamic model predicted the times at which the land-fast ice started to break-up accurately to within 4.7
hours to up to almost six days (Figure 18 and Table 4). However, these model results assumed a uniform snow cover



of 0.4 m on the sea ice on April 1 for the whole model domain of coastal Labrador. Given the extreme uncertainty of
this initial condition, it is imperative that operational use of this model utilize accurate measurements of snow cover
for the initialization of the model. It may be possible to hindcast initial snow depths by fitting the thermodynamic
model to seasonal evolutions of ice cover as documented by the CIS ice charts.
The thermodynamic model can be used to generate seasonal forecasts of the timing and spatial pattern of the break-
up of the land-fast ice along coastal Labrador. Such forecasts are important to operational planning for possible
future oil and gas exploration and production operations in this region, as well as vessel transport operations such as
those associated with the Voisey's Bay nickel mine. Access to ports near Makkovik and Nain is not possible for low
or non-ice class vessels prior to the break-up of the land-fast ice.
The coupled thermodynamic-dynamic model can be used to issue forecasts of the evolution of the coastal Labrador
ice cover over shorter periods of days to weeks compared to the thermodynamic-only model, which can help inform
the planning of ice management operations and vessel routing. The modeled ice drift speeds during May 1-7, 2015
were within 1.5 m s$^{-1}$ of those measured by the tracking buoys deployed on the land-fast ice during April (Figure
31). Given the fact that the dynamic model does not account for ice entering or leaving the edges of the model
domain, the model should be run for as large an area as possible and for shorter periods, with the selected model
domain boundaries far away from particular locations of interest for ice forecasting. The temporal and spatial
resolution of the model can be adjusted by the user, with computational efficiency and model runtimes improving at
lower resolutions.
The model is most likely sensitive to a number of the thermodynamic and dynamic parameters such as the number
of nearest-neighbor ice particles and the smoothing length used in the SPH scheme, among others. Further study of
the model's sensitivity to these parameters is warranted. In addition, the thermodynamic model for prediction of the
onset of land-fast ice break-up does not account for the effects of wind stress. The wind speed in the current
thermodynamic model only affects the ice through the sensible and latent heat fluxes (Eqs. 32 and 39, respectively).
Future development of this model should account for the tensile stresses imposed on the offshore land-fast ice
boundary by the offshore component of the wind.
Finally, it is important to note that the results shown in this paper were generated with observed and hindcast
metocean data. Operational use of this model will require forecast metocean data, which will increase the errors in
the ice forecasts. However, the model can be reinitialized in near-real-time with the most up-to-date locally observed
gridded ice and metocean datasets in order to improve regional ice forecasts. Export of this model to other ice
environments is also an important aspect of this model development effort.
**Code and Data Availability**
Upon request to the lead author, the ice buoy drift data and model output raw data can be provided. The model code
is proprietary to C-CORE; however, for the review process, the model code can be made available temporarily to the

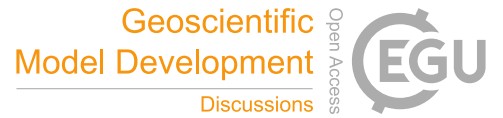



Ebert, E.E., and Curry, J.A. (1993), An Intermediate One-Dimensional Thermodynamic Sea Ice Model for
Investigating Ice-Atmosphere Interactions, Journal of Geophysical Research, 98(C6), 10,085-10,109.

European Center for Medium-Range Weather Forecasting (ECMWF), ERA-Interim Reanalysis, Accessed on the
World Wide Web: http://apps.ecmwf.int/datasets/data/interim-full-daily/levtype=sfc/, March 8, 2016.

Food and Agriculture Organization (FAO) of the United Nations (2015), Chapter 3 – Meteorological data. Accessed
on the World Wide Web: http://www.fao.org/docrep/x0490e/x0490e07.htm, March 8, 2016.

Fenty, I., and Heimbach, P. (2013), Coupled Sea-Ice-Ocean-State Estimation in the Labrador Sea and Baffin Bay,
Journal of Physical Oceanography, 43(5), 884-904.

Fofonoff, N.P., and Millard Jr., R.C. (1983), Algorithms for computation of fundamental properties of seawater,
Unesco technical papers in marine science, 44.

General Bathymetric Chart of the Oceans (GEBCO) (2008), Accessed on the World Wide Web:
http://www.gebco.net/data_and_products/gridded_bathymetry_data/, March 8, 2016.

Global Ocean 1/12° Physics Analysis and Forecast, Accessed on the World Wide Web:
http://marine.copernicus.eu/web/69-interactive-
catalogue.php?option=com_csw&view=details&product_id=GLOBAL_ANALYSIS_FORECAST_PHYS_
001_002, March 8, 2016.

Gutfraind, R., and Savage, S.B. (1997), Smoothed Particle Hydrodynamics for the Simulation of Broken-Ice Fields:
Mohr-Coulomb-Type Rheology and Frictional Boundary Conditions, Journal of Computational Physics,
134, 203-215.

Hibler III, W.D. (1979), A Dynamic Thermodynamic Sea Ice Model, Journal of Physical Oceanography, 9, 815-846.
Holland, M.M., Bailey, D.A., Briegleb, B.P., Light, B., and Hunke, E. (2012), Improved Sea Ice Shortwave
Radiation Physics in CCSM4: The Impact of Melt Ponds and Aerosols on Arctic Sea Ice, Journal of
Climate, 25, 1,413-1,430.

International Standards Organization Draft International Standard (ISO/DIS 19901-1) (2013), Petroleum and natural
gas industries – Specific requirements for offshore structures – Part 1: Metocean design and operating
considerations.

Khandekar, M.L. (1980), Inertial oscillations in floe motion over the Beaufort Sea – observations and analysis,
Atmosphere-Ocean, 18(1), 1-14.

Konig-Langlo, G., and Augstein, E. (1994), Parameterization of the downward long-wave radiation at the Earth's
surface in polar regions, Meteorologische zeitschrift, N.F.3, Jg. 1994, H.6, 343-347.

Lazier, J.R.N., and Wright, D.G. (1993), Annual Velocity Variations in the Labrador Current, Journal of Physical
Oceanography, 23, 659-678.



Lindsay, R.W., and Stern, H.L. (2004), A New Lagrangian Model of Arctic Sea Ice, Journal of Physical
Oceanography, 34, 272-283.

MacKinnon, A. (2015), The Dufort-Frankel Method. Accessed on the World Wide Web:
http://www.cmth.ph.ic.ac.uk/people/a.mackinnon/Lectures/compphys/node35.html, March 8, 2016.

Matthews, J.H., and Fink, K.K. (2004), Numerical methods using Matlab, 4th Edition, Prentice-Hall, Inc., Upper
Saddle River, New Jersey, USA 0-13-065248-2, 505-508.

Maykut, G.A., and Untersteiner, N. (1971), Some results from a time dependent thermodynamic model of sea ice,
Journal of Geophysical Research, 76, 1,550-1,575.

Mitchell, A.R., and Griffiths, D.F. (1980), The Finite Difference Method in Partial Differential Equations, John
Wiley, New York, USA.

National Oceanographic and Atmospheric Administration (NOAA) (2015), General Solar Position Calculations.
Accessed on the World Wide Web: http://www.esrl.noaa.gov/gmd/grad/solcalc/solareqns.PDF, March 8,
2016.

North American Regional Reanalysis (NARR), NCEP Reanalysis data provided by the NOAA/OAR/ESRL PSD,
Boulder, Colorado, USA, from their Web site at http://www.esrl.noaa.gov/psd/, March 8, 2016.

Peng, S. (1989), Application of a Simple Ice-Ocean Model to the Labrador Sea. Master of Science Thesis,
Department of Meteorology, McGill University, Montreal, Quebec, Canada.

Prinsenberg, S.J., Peterson, I.K., Holladay, J.S., and Lalumiere, L. (2011), Snow and Ice Thickness Properties of
Lake Melville, a Canadian Fjord Located along the Labrador Coast, Proceedings of the Twenty-first (2011)
International Offshore and Polar Engineering Conference (ISOPE), Maui, Hawaii, USA, June 19-24, 2011.

Prinsenberg, S.J., and Yao, Q. (1999), The Seasonal Evolution of Sea Ice Cover off eastern Canada as Simulated by
a Coupled Ice-Ocean Model for 1991/92 and Expected 2×CO$_2$ atmospheric conditions. Accessed on the
World Wide Web: http://www.bio.gc.ca/science/research-recherche/ocean/ice-
glace/documents/prins05.pdf, March 8, 2016.

Sayed, M., Carrieres, T., Tran, H., and Savage, S.B. (2002), Development of an Operational Ice Dynamics Model
for the Canadian Ice Service, Proceedings of the Twelfth (2002) International Offshore and Polar
Engineering Conference (ISOPE), Kitakyushu, Japan, May 26-31, 2002.

Timco, G.W., and Frederking, R.M.W. (1990), Compressive Strength of Sea Ice Sheets, Cold Regions Science and
Technology, 17, 227-240.

Timco, G.W., and Johnston, M.E. (2002), Sea Ice Strength during the Melt Season. Proceedings of the 16th
International Association of Hydraulic Engineering and Research (IAHR) International Symposium on Ice,
Dunedin, New Zealand, December 2-6, 2002.

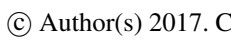



Tonboe, R.T., Dybkjaer, G., and Hoyer, J.L. (2011), Simulations of the snow covered sea ice surface temperature
and microwave effective temperature, Tellus, 63A, 1,028-1,037.

Tsonis, A. (2002), An Introduction to Atmospheric Thermodynamics. Cambridge University Press 0-521-79676-8,
182.

Vale        (2015),        Voisey's        Bay.        Accessed        on        the        World        Wide        Web:
http://www.vale.com/canada/EN/business/mining/nickel/vale-canada/voiseys-bay/Pages/default.aspx,
March 8, 2016.

Water equivalent to snow ratios: DOC/NOAA/NWS (1996), Observing Handbook Number 7, Surface Observations
part IV, Supplementary Observations, Table 2-14, 440 pp.

Woods    Hole    Oceanographic    Institute    (2010),    Humidity,    Accessed    on    the    World    Wide    Web:
http://www.whoi.edu/page.do?pid=30578, March 8, 2016.

Yao, T. (2000), Assimilating Sea Surface Temperature Data into an Ice-Ocean Model of the Labrador Sea, Canadian
Technical Report of Hydrography and Ocean Sciences, 212.

Yao, T., Tang, C.L., Carrieres, T., and Tran, D.H. (2000a), Verification of a coupled ice ocean forecasting system
for the Newfoundland shelf, Atmosphere-Ocean, 38(4), 557-575.

Yao, T., Tang, C.L., and Peterson, I.K. (2000b), Modeling the seasonal variation of sea ice in the Labrador Sea with
a coupled multicategory ice model and the Princeton ocean model, Journal of Geophysical Research,
105(C1), 1,153-1,165.