# Peer review of "An Operational Thermodynamic-Dynamic Model for the Coastal 1"

_Geoscientific Model Development, 2017_

## Short Comment (SC1) · 4 Apr 2017

Dear authors,

in my role as Executive editor of GMD, I would like to bring to your attention our Editorial version 1.1:

http://www.geosci-model-dev.net/8/3487/2015/gmd-8-3487-2015.html

This highlights some requirements of papers published in GMD, which is also available on the GMD website in the 'Manuscript Types' section:

http://www.geoscientific-model-development.net/submission/manuscript_types.html

In particular, please note that for your paper, the following requirements have not been met in the Discussions paper:

- "The main paper must give the model name and version number (or other unique identifier) in the title."

- "If the model development relates to a single model then the model name and the version number must be included in the title of the paper. If the main intention of an article is to make a general (i.e. model independent) statement about the usefulness of a new development, but the usefulness is shown with the help of one specific model, the model name and version number must be stated in the title. The title could have a form such as, "Title outlining amazing generic advance: a case study with Model XXX (version Y)"."

In order to simplify reference to your developments, please add a model name (and/or its acronym) and a version number in the title of your article in your revised submission to GMD.

Yours,

Astrid Kerkweg

––––––––––––––––––––––––––––––––––

---

## Author Comment (AC1) · 4 Apr 2017

The comment was uploaded in the form of a supplement:
http://www.geosci-model-dev-discuss.net/gmd-2017-39/gmd-2017-39-AC1-supplement.zip

---

## Referee Comment (RC1) · Anonymous Referee #1 · 10 Apr 2017

The paper describes a model for predicting the sea ice break up near the coast. The skill in predicting such events is important for offshore industries operating in cold regions. I have two major concerns to this paper: justifications of developing this particular model and reproducibility of results.

There exist a few open source communities for sea ice modeling, e.g. CICE and LIM for a standalone sea-ice model and MITgcm for coupled ocean and sea ice. The manuscript does not address potential deficiencies of these tools in predicting the sea-ice break up, which should be the motivation for developing a new model. It is also unclear that the model presented in the manuscript performs better than these existing tools. In the present form, the paper does not articulate the advantages or needs of

creating a new model for this application.

The model is largely based on the published one-dimensional thermodynamic sea-ice model, and the paper does not contain new formulations of physics or adaptations of innovative methods to solve the governing equations. The model ability to predict the break up is sensitive to the initial snow depth over the ice. This raises a doubt in applying this tool as a generic forecasting platform, and the authors do not discuss a possible mitigation strategy to constrain the uncertainties. Furthermore, the model code will not be shared with readers, which questions the reproducibility of the results in the future. In summary, I do not see that this manuscript fits the goal of the journal in the present form, and therefore, I recommend rejection at this stage.

––––––––––––––––––––––––––––––––

---

## Referee Comment (RC2) · Anonymous Referee #2 · 23 May 2017

The paper describes a thermodynamic-dynamic model of sea ice in the coastal area. The intention of the model is to predict the break-up of sea ice and its drift for operational use. The motivation and need of such a model are understandable and probably of a high demand for many applications in different geographical areas, not only in the one described in the paper. Thus, authors' effort in developing such a model is valuable, but it is clear that at the moment this is the first iteration in model development, which is hard to validate yet and implement into operational use.

"The thermodynamic model runs use a 0.45° spatial resolution (approximately 50 km) for initial ice particle spacing and a one-hourly temporal resolution, while the thermodynamic-dynamic model run uses a 0.5° spatial resolution (approximately 55.5

km) for initial ice particle spacing and a one-minute temporal resolution." "The thermodynamic model can be used to generate seasonal forecasts of the timing and spatial pattern of the break-up of the land-fast ice along coastal Labrador" "The coupled thermodynamic-dynamic model can be used to issue forecasts of the evolution of the coastal Labrador ice cover over shorter periods of days to weeks compared to the thermodynamic-only model, which can help inform the planning of ice management operations and vessel routing."

The authors themselves stress the points which introduce the highest uncertainties: a) in thermodynamic model it is snow thickness distribution over calculation domain at the moment of model initialization; b) in dynamic model it is a boundary condition preventing ice enter and leave the calculation domain; c) model does not account for the effects of wind stress. Another point which is mentioned briefly by authors and seen during the review process is that there is a systematic difference for model prediction for two different regions (Table 4 and Figure 27 – Figure 30). As it follows from results (Figure 19, Figure 27 – Figure 31), authors conclude that model is better at simulating processes further offshore. In my opinion, it has to be related to the spatial resolution of the calculation domain, which has to be increased for nearshore areas.

Physical processes are modeled based on the previously existing models, and there are many parameters (snow compaction (neglected here), drag coefficients, albedo coefficients, internal friction angle...) which has to be assumed or chosen and which can be tuned to affect the results of the model simulations. Further, I have a comment regarding section 2, "Ice Environmental Modelling." Since the models described there are previously existed models, I would recommend combining only subsections 2.n and all subsections 2.n.m move to an appendix. Or even more radically, remove it at all from the paper, leaving references to the previous papers, since most of them are reproduced. Instead, I would like to see a more wide explanation about the model and even a scheme of the computational process where a reader could see how different physical modules are coupled between each other. The model is the focus of the work

and, and as for now, section 2 looks more like a technical documentation of the model, which makes it a bit difficult to follow the manuscript line. Though I support including that descriptive part into the appendix.

At the moment hindcast data of ocean and atmosphere were supplied into the model, which is complexing the validation process. For further model validation runs, forecasted data should be used. Definition of ice break up criteria should be formulated and explained more thorough.

Otherwise, the manuscript is written in a well-structured manner and convenient enough to follow, besides few comments which will be given below in line by line comments.

Overall, after - restructuring Section 2 with a focus on describing the structure of the model instead of describing previously existing physical models, - introducing a scheme of the computational process, - adding discussion on validation of the model and breakup criteria, - and addressing line by line comments given below, I could recommend this manuscript to be accepted for publication, understanding that this is first results of the developing model and further development of the model is needed

70 Beginning from the Figure 1 and further, especially in the results section, I recommend to mark Makkovik and Nain on all maps by text in addition to signs. This will make it easier to follow the manuscript, as it often refers to these locations.

74 is it reliable enough technique to determine on ice drift periods? I assume you see different level temperature or fewer fluctuations, when in water?

78 I would strongly recommend redefining naming system of buoys. I believe it is hard for you to deflect from the original system, but would be very beneficial for the manuscript and readers. For example when you present results in Figure 27 – Figure 30 it is confusing to understand the results when you see first Buoy 1 (from Nain), then Buoy 2 (from Makkovik) and Buoy 3 (from Makkovik) and then Buoy 4 (again

from Nain). Consider to change it e.g. to Buoys 1-3 (from Nain) and Buoys 4-6 (from Makkovik) or even Buoys N1, N2, N3 and M1, M2, M3 respectively?

101, 107 Expect that Figure 3 and Figure 4 will be submitted of a better quality than presented now.

136 "In this paper,... " In your paper or in Yao et al., 2000b ?

283 "One melt ponds..." Seems there is a typo: Once melt ponds...

389, 397 As mentioned above in overview section, Table 2 and Table 3 are reproduced from EC93. Consider to remove them or move to the appendix.

431, 439 consider to explain/refer why air and water drag coefficients are chosen of such values?

506 consider to explain/refer why coastal cells radius is chosen of such particular size and water drag coefficients are chosen of such values?

647 Is it so that after April 23rd CIS-LIM2 Ice Thickness Normalized Error has changed sign? Since ice concentration and thickness graphs presented for all weeks, some discussion of them is expected.

702, 709 replace green sign by blue to be consistent with Figure 1

730 consider changing magenta color to avoid mixing with red color

741 "..., it can reasonably predict the progression and timing of the decline in regional ice concentration and thickness more than a month in advance." Since hindcasted metocean data were used and not forecasted, the above one songs questionable.

748 how choice of the concentration threshold (0.99) would affect times 4.7 hours and 5.9 days

753 Figure 18a is not complete/consistent to Figure 18b, Figure 18c and Figure 18d. Few lines are missing

781 for comparison with the original state of ice concentration and ice thickness, consider adding at the end of paragraph something in line with the following: ..., compare to its initial state (Figure 16)

831 Figure 26b,d. What is the reason for such high difference in thickness in results?

846 Figure 27 – Figure 30. It is clear that different regions give different estimates. Combining figures by regions could be helpful. First Figure 27, Figure 30 and then Figure 28, Figure 29.

870 Here, dates (May 3, 5, 7) for which results are shown in Figure 32 – Figure 34 are not consistent with Table 5 and line 777 (May 2, 3, 7). The explanation of May 2, 3, 7 choice was that buoys started to move 1, 2 and 6 respectively.

890 Here, dates (May 3, 5, 7) for which results are shown in Table 6 are not consistent with Table 5 and line 777 (May 2, 3, 7). The explanation of May 2, 3, 7 choice was that buoys started to move 1, 2 and 6 respectively.